# Optimal Unbiased Randomizers for Regression with Label Differential Privacy

**Ashwinkumar Badanidiyuru**
Google
Mountain View, CA

**Badih Ghazi**
Google Research
Mountain View, CA

**Pritish Kamath**
Google Research
Mountain View, CA

**Ravi Kumar**
Google Research
Mountain View, CA

**Ethan Leeman**
Google
Cambridge, MA

**Pasin Manurangsi**
Google Research
Bangkok, Thailand

**Avinash V Varadarajan**
Google
Mountain View, CA

**Chiyuan Zhang**
Google Research
Mountain View, CA

## Abstract

We propose a new family of label randomizers for training *regression* models under the constraint of label differential privacy (DP). In particular, we leverage the trade-offs between bias and variance to construct better label randomizers depending on a privately estimated prior distribution over the labels. We demonstrate that these randomizers achieve state-of-the-art privacy-utility trade-offs on several datasets, highlighting the importance of reducing bias when training neural networks with label DP. We also provide theoretical results shedding light on the structural properties of the optimal unbiased randomizers.

## 1 Introduction

Differential privacy (DP) [DMNS06] has gained significant importance in recent years as a mathematically sound metric for rigorously quantifying the potential disclosure of personal user information through ML models [ACG+16, RE19, TM20]. DP guarantees that the model generated by the training process remains statistically indistinguishable, even if the data contributed by any individual user to the training dataset is modified arbitrarily.

In certain scenarios, the training *features* of a specific example are already accessible to potential adversaries, while only the training *label* is considered sensitive. For instance, within the context of digital advertising, it is typical to train conversion models that aim to predict whether a user, who interacts with an advertisement on a publisher's website, will ultimately make a purchase of the advertised item on the advertiser's website. The conversion label, which indicates whether the user completed the purchase or not, is not initially known to the publisher website and is considered sensitive information that spans multiple websites.[1] This particular setting, referred to as label DP, was initially studied in the work of [CH11]. Subsequently, it has garnered attention in various recent works, such as [GGK+21, MEMP+21, EMSV22][2].

For regression objectives (such as the squared loss and the Poisson log loss), the state-of-the-art is given by the prior work of [GKK+23]. Their algorithm works in the so-called *features-oblivious* setting (Figure 1), which consists of a *features party* that has access to all the features, and a *labels party* that has access to all the labels. The labels party applies a mechanism that is DP with respect to the labels in order to produce the message $M(\cdot)$ that it sends to the features party, who then uses the

---

[1]Given the announcement of various web browsers and mobile platforms regarding the deprecation of third-party cookies, which were previously employed to track user behavior across different websites, the need for training models that protect the privacy of labels has become increasingly important.

[2]as well as in proposals for industry-wide web and mobile standards [Har23].

37th Conference on Neural Information Processing Systems (NeurIPS 2023).

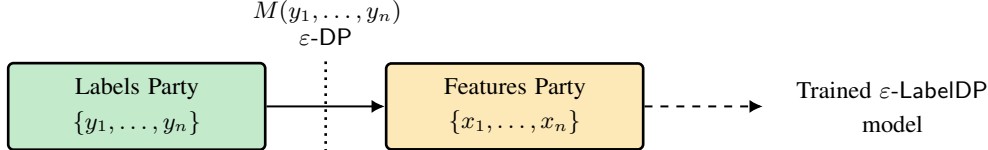

Figure 1: Feature-oblivious label DP model training studied in [GKK$^+$23].

features along with the incoming message in order to train the model (which as a consequence would also satisfy label DP). It proceeds by using part of the privacy budget in order to estimate a prior over the labels, and then constructs a randomizer optimizing the "noisy label loss" (see Definition 4) with respect to this prior.

In this work, we introduce a novel mechanism with bias-limiting constraints, motivated by the theory of bias-variance trade-offs. We show that while these constraints lead to significantly higher noisy label loss, the models trained on the privatized labels performs surprisingly well, achieving the state-of-the-art utility-privacy trade-off on multiple real world datasets.

**Organization.** Section 2 provides some background definitions related to (label-)DP and learning. In Section 3, we present the new label DP randomizers obtained by imposing unbiasedness constraints. Section 4 provides an experimental evaluation demonstrating that our method achieves state-of-the-art privacy-utility trade-offs across multiple datasets. We provide some related work in Section 5. We conclude with some future directions in Section 6. All missing proofs along with additional related work, as well as details on the experimental evaluation are provided in the Appendix.

## 2 Preliminaries

Let $\mathcal{D}$ be an unknown distribution on $\mathcal{X} \times \mathcal{Y}$. We consider the regression setting where $\mathcal{Y} \subseteq \mathbb{R}$; let $\mathcal{P}$ denote the marginal distribution of $\mathcal{D}$ on $\mathcal{Y}$. In supervised learning, we have a set $\{(x, y)\}$ of examples drawn from $\mathcal{D}$. The goal is to learn a predictor $f_\theta$ to minimize the expected loss $\mathcal{L}_\mathcal{D}(f_\theta) := \mathbb{E}_{(x,y) \sim \mathcal{D}} \ell(f_\theta(x), y)$, for some loss function $\ell : \mathbb{R} \times \mathcal{Y} \to \mathbb{R}$. Some commonly-used loss functions used in regression tasks are the *Poisson log loss* $\ell_{\mathrm{Poi}}(\tilde{y}, y) := \tilde{y} - y \cdot \log(\tilde{y})$ and the *squared loss* $\ell_{\mathrm{sq}}(\tilde{y}, y) := \frac{1}{2}(\tilde{y} - y)^2$.

We recall the definition of DP; for more background, see the book of Dwork and Roth [DR14].

**Definition 1** (DP; [DMNS06])**.** Let $\varepsilon \in \mathbb{R}_{>0}$. A randomized algorithm $\mathcal{A}$ is said to be $\varepsilon$-*differentially private* (denoted $\varepsilon$-DP) if for any two "adjacent" input datasets $X$ and $X'$, and any measurable subset $S$ of outputs of $\mathcal{A}$, it holds that $\Pr[\mathcal{A}(X) \in S] \leq e^\varepsilon \cdot \Pr[\mathcal{A}(X') \in S]$.

In the context of supervised learning, an algorithm produces a model as its output, while the labeled training set serves as the input. Two inputs are considered "adjacent" if they differ on a single training example. This concept of adjacency is intended to safeguard both the features and the label of any individual example. However, in certain scenarios, protecting the features may either be unnecessary or infeasible, and the focus is solely on protecting the labels. This leads to the following definition:

**Definition 2** (Label DP; [CH11])**.** A randomized training algorithm $\mathcal{A}$ is said to be $\varepsilon$-*label differentially private* (denoted $\varepsilon$-LabelDP) if it is $\varepsilon$-DP when two input datasets are "adjacent" if they differ on the label of a single training example.

We recall the notion of feature-oblivious label DP [GKK$^+$23]. In this scenario, there are two parties: the *features party* and the *labels party*. The features party has the sequence $(x_i)_{i=1}^n$ of all feature vectors across the $n$ users; the labels party has the sequence $(y_i)_{i=1}^n$ of the corresponding labels. The labels party sends a single (possibly randomized) message $M(y_1, \ldots, y_n)$ to the features party. Based on its input and the received message, the features party generates an ML model as its output.

**Definition 3** (Feature-Oblivious Label DP [GKK$^+$23])**.** In the above scenario, the output of the features party satisfies *feature-oblivious $\varepsilon$-LabelDP* if the message $M(y_1, \ldots, y_n)$ is $\varepsilon$-DP where two inputs are considered "adjacent" if they differ on a single $y_i$.

## 3 Label-DP Randomizers

A standard recipe for learning with label DP is to: (i) compute noisy labels using a local DP randomizer $\mathcal{M}$ and (ii) use a learning algorithm on the dataset with noisy labels. Many of the baseline algorithms that we consider follow this recipe, through different ways of generating noisy labels, such as, (a) (continuous/discrete) Laplace mechanism, (b) (continuous/discrete) staircase mechanism, etc. (see Appendix F for formal definitions). Prior work by [GKK$^+$23] argues that intuitively such a learning algorithm will be most effective when the noisy label "mostly agrees" with the true label. This was formalized in the goal of minimizing the *noisy label loss*, which is the expected loss between the true label and the noisy label, for the true label drawn from a prior distribution, for some loss function $\ell$.

**Definition 4** (Noisy label loss). For a loss function $g : \mathbb{R} \times \mathcal{Y} \to \mathbb{R}$, the *noisy label loss* of a randomizer $\mathcal{M}$ with respect to prior $\mathcal{P}$ is $\mathcal{G}(\mathcal{M}; \mathcal{P}) := \mathbb{E}_{y \sim \mathcal{P}, \hat{y} \sim \mathcal{M}(y)} \, g(\hat{y}, y)$, where $\hat{y}$ is the noisy label generated by $\mathcal{M}$ on input $y$.

This was motivated by the triangle inequality [GKK$^+$23, Eq. (1)], namely,

$$\mathop{\mathbb{E}}_{(x,y)\sim\mathcal{D}} \ell(f_\theta(x), y) \;\leq\; \mathop{\mathbb{E}}_{\substack{y\sim\mathcal{P}\\\hat{y}\sim\mathcal{M}(y)}} \ell(\hat{y}, y) \;+\; \mathop{\mathbb{E}}_{\substack{(x,y)\sim\mathcal{D}\\\hat{y}\sim\mathcal{M}(y)}} \ell(f_\theta(x), \hat{y}) \,. \tag{1}$$

The work of [GKK$^+$23] focused on optimizing noisy label loss; they showed that the optimal randomizer takes the form of "Randomized-Response on Bins" (see Appendix F for more details). However, if we notice carefully, the noisy label loss does not depend on the number of examples, so the RHS of (1) does not go to zero as the number of examples goes to infinity, even though we would like the excess population loss of the learnt predictor to asymptotically go to zero. In this paper, we provide an alternative insight into the goal of minimizing the noisy label loss.

For any distribution $\mathcal{D}$ over $\mathcal{X} \times \mathcal{Y}$, and any label randomizer $\mathcal{M}$ mapping $\mathcal{Y}$ to $\hat{\mathcal{Y}} \subseteq \mathbb{R}$, let $\mathcal{D}_{\mathcal{M}}$ be the distribution over $\mathcal{X} \times \hat{\mathcal{Y}}$ sampled as $(x, \mathcal{M}(y))$ for $(x, y) \sim \mathcal{D}$. The *Bayes optimal predictor* for any distribution $\mathcal{D}$ over $\mathcal{X} \times \mathcal{Y}$ w.r.t. loss $\ell$, is a predictor $f_{\mathcal{D}}^* : \mathcal{X} \to \mathbb{R}$ that minimizes $\mathcal{L}_{\mathcal{D}}(f)$, which roughly corresponds to the best predictor we could hope to learn with unbounded number of samples from $\mathcal{D}$. We show that, when training with a loss from a broad family that includes $\ell_{\mathrm{sq}}$ and $\ell_{\mathrm{Poi}}$, the Bayes optimal predictor is preserved after applying the label randomizer iff $\mathbb{E}[\mathcal{M}(y)] = y$ holds for all $y \in \mathcal{Y}$ (i.e., the randomizer is *unbiased*).

**Theorem 5.** *Suppose $\ell$ is a loss function such that for all distributions $\mathcal{P}$ over $\mathcal{Y}$, the minimizer $\hat{y}_* := \min_{\hat{y}\in\mathbb{R}} \mathbb{E}_{y\sim\mathcal{P}} \, \ell(\hat{y}, y)$ exists and is given as $\hat{y}_* = \mathbb{E}_{y\sim\mathcal{P}}[y]$. Then the following are equivalent:*

▷ $\mathbb{E}[\mathcal{M}(y)] = y$ *holds for all* $y \in \mathcal{Y}$,
▷ *For all distributions $\mathcal{D}$ over $\mathcal{X} \times \mathcal{Y}$, it holds that $f_{\mathcal{D}}^* = f_{\mathcal{D}_{\mathcal{M}}}^*$.*

The main observation is that $f_{\mathcal{D}}^*(x_0) = \mathbb{E}[y \mid x = x_0]$ with probability 1 over $x_0$, and similarly, $f_{\mathcal{D}_{\mathcal{M}}}^*(x_0) = \mathbb{E}[\mathcal{M}(y) \mid x = x_0]$ with probability 1 over $x_0$. We defer the full proof to Appendix A, but for now we demonstrate a simple example where a predictor learned using noisy labels produced by the optimal RR-on-Bins randomizer can in fact be sub-optimal. Let $\mathcal{X} = \{\mathtt{a}, \mathtt{b}\}$ and $\mathcal{Y} = \{0, 1, 2\}$ and the distribution $\mathcal{D}$ be defined in Table 1.

|   | 0 | 1 | 2 |
|---|---|---|---|
| a | 0.35 | 0.1 | 0.05 |
| b | 0.25 | 0.15 | 0.1 |

Table 1: Example $\mathcal{D}$

The Bayes optimal predictor for $\mathcal{D}$ w.r.t. $\ell_{\mathrm{sq}}$ (or even $\ell_{\mathrm{Poi}}$) is given as $f_{\mathcal{D}}^*(\mathtt{a}) = 0.4$ and $f_{\mathcal{D}}^*(\mathtt{b}) = 0.7$. On the other hand, the optimal $\mathcal{M} = $ RR-on-Bins$_\varepsilon^\Phi$ randomizer for $\varepsilon = 0.5$ (see Appendix F for notation), corresponds to the map $\Phi(0) \approx 0.396$ and $\Phi(1) = \Phi(2) \approx 0.720$. The Bayes optimal predictor for $\mathcal{D}_{\mathcal{M}}$ is given as $f_{\mathcal{D}_{\mathcal{M}}}^*(\mathtt{a}) = 0.542$ and $f_{\mathcal{D}_{\mathcal{M}}}^*(\mathtt{b}) = 0.558$. The squared loss of these predictors are given as $\mathcal{L}_{\mathcal{D}}(f_{\mathcal{D}}^*) = 2.625$ and $\mathcal{L}_{\mathcal{D}}(f_{\mathcal{D}_{\mathcal{M}}}^*) = 2.726$, the latter being sub-optimal.

Thus, preserving the Bayes optimal predictor is a desirable property of a label randomizer, as any learning algorithm that converges to the optimal predictor can also be trained on the noisy labels and will approach the predictor for the original distribution.

Additionally, we can relate the property $\mathbb{E}[\mathcal{M}(y)] = y$ to the unbiasedness of the gradients obtained in SGD. First, we observe that the gradient with respect to the parameters for $\ell_{\mathrm{sq}}$ and $\ell_{\mathrm{Poi}}$ are affine in $y$, namely,

$$\nabla_\theta \ell_{\mathrm{sq}}(f_\theta(x), y) = (f_\theta(x) - y) \cdot \nabla_\theta f_\theta(x) \,, \qquad \nabla_\theta \ell_{\mathrm{Poi}}(f_\theta(x), y) = (1 - y/f_\theta(x)) \cdot \nabla_\theta f_\theta(x) \,.$$

Thus, the error in the gradient estimate when using the noisy label $\hat{y}$ instead of $y$ is given as

$$\nabla_\theta \ell_{\mathrm{sq}}(f_\theta(x), \hat{y}) - \nabla_\theta \ell_{\mathrm{sq}}(f_\theta(x), y) \;=\; (y - \hat{y}) \cdot \nabla_\theta f_\theta(x),$$

$$\nabla_\theta \ell_{\mathrm{Poi}}(f_\theta(x), \hat{y}) - \nabla_\theta \ell_{\mathrm{Poi}}(f_\theta(x), y) \;=\; (y - \hat{y}) \cdot \frac{\nabla_\theta f_\theta(x)}{f_\theta(x)}.$$

Let $S = \{(x_1, y_1), \ldots, (x_b, y_b)\}$ be a mini-batch of examples and let $\hat{S} = \{(x_1, \hat{y}_1), \ldots, (x_b, \hat{y}_b)\}$ be the mini-batch with noisy labels as returned by the randomizer $\mathcal{M}$. Also consider the dataset with expected noisy labels $\tilde{S} = \{(x_1, \tilde{y}_1), \ldots, (x_b, \tilde{y}_b)\}$, where $\tilde{y}_i = \mathbb{E}_{\hat{y} \sim \mathcal{M}(y)} \hat{y}$. The difference between a mini-batch gradient w.r.t. noisy labels and the gradient of the population loss decomposes for $\ell_{\mathrm{sq}}$ as follows (similar decomposition holds for $\ell_{\mathrm{Poi}}$):

$$
\begin{aligned}
&\nabla_\theta \mathcal{L}_{\hat{S}}(f_\theta) - \nabla_\theta \mathcal{L}_{\mathcal{D}}(f_\theta) \\
&= \nabla_\theta \mathcal{L}_S(f_\theta) - \nabla_\theta \mathcal{L}_{\mathcal{D}}(f_\theta) + \nabla_\theta \mathcal{L}_{\tilde{S}}(f_\theta) - \nabla_\theta \mathcal{L}_S(f_\theta) + \nabla_\theta \mathcal{L}_{\hat{S}}(f_\theta) - \nabla_\theta \mathcal{L}_{\tilde{S}}(f_\theta) \qquad (2)\\
&= \underbrace{\nabla_\theta \mathcal{L}_S(f_\theta) - \nabla_\theta \mathcal{L}_{\mathcal{D}}(f_\theta)}_{(a)} + \underbrace{\mathop{\mathbb{E}}_{(x,y)\in S}(y - \mathbb{E}\,\hat{y}) \cdot \nabla_\theta f_\theta(x)}_{(b)} + \underbrace{\mathop{\mathbb{E}}_{(x,y)\in S}(\mathbb{E}\,\hat{y} - \hat{y}) \cdot \nabla_\theta f_\theta(x)}_{(c)}.
\end{aligned}
$$

The term $(a)$ is the difference between the mini-batch gradient and the population gradient, the inherent stochasticity in mini-batch SGD, which has zero bias. The terms $(b)$ and $(c)$ form precisely the bias-variance decomposition for the additional stochasticity in the gradient introduced by the label noise, with the term $(c)$ having zero bias. In the case of stochastic convex optimization, it is well known that with access to biased gradients, it is impossible to achieve vanishingly small excess loss; whereas, with access to gradients with zero bias and any finite variance, it is possible to achieve an arbitrarily small excess loss using sufficient number of steps of stochastic gradient descent (see Appendix D for more details). Motivated by this reasoning, our approach is to use a randomizer that has the *smallest variance possible while ensuring zero bias*, namely the term $(b)$ is zero.

### 3.1 Computing Optimal Unbiased Randomizers

We consider a label randomizer that minimizes the noisy label loss, while being unbiased. Namely, we say that an $\varepsilon$-DP randomizer $\mathcal{M}$ that maps the label set $\mathcal{Y} \subseteq \mathbb{R}$ to $\mathbb{R}$ is *unbiased* if it satisfies $\mathbb{E}_{\hat{y} \sim \mathcal{M}(y)} \hat{y} = y$ for all input labels $y \in \mathcal{Y}$. We use a linear programming (LP) based algorithm (Algorithm 1) to compute an unbiased randomizer that minimizes the noisy label loss. In order to keep this approach computationally tractable we require that both $\mathcal{Y}$ and $\hat{\mathcal{Y}}$ are finite; as we discuss shortly, it is possible to handle general $\mathcal{Y}$ by discretization using randomized rounding, and moreover the excess noisy label loss due to consideration of a discrete $\hat{\mathcal{Y}}$ can be bounded as well. Even though Algorithm 1 is general, we will only consider $g = \ell_{\mathrm{sq}}$ henceforth (irrespective of the training loss $\ell$).

---

**Algorithm 1** ComputeOptUnbiasedRand$_\varepsilon$

---

**Parameters:** Privacy parameter $\varepsilon \geq 0$.
**Input:** $\mathcal{P} = (p_y)_{y \in \mathcal{Y}}$—prior over input labels $\mathcal{Y}$,
$\qquad\;\; \hat{\mathcal{Y}} = (\hat{y}_i)_{i \in \mathcal{I}}$—a finite sequence of potential output labels.
**Output:** An $\varepsilon$-DP label randomizer.

Solve the following LP in variables $M = (M_{y \to i})_{y \in \mathcal{Y}, i \in \mathcal{I}}$:

$$
\begin{aligned}
\min_M \quad & \textstyle\sum_{y \in \mathcal{Y}} p_y \left( \sum_{\hat{y} \in \hat{y}} M_{y \to i} \cdot g(\hat{y}_i, y) \right), \quad \text{subject to} \\
[\text{Non-negativity}] \quad & \forall y \in \mathcal{Y},\, i \in \mathcal{I}: && M_{y \to i} \geq 0 \\
[\text{Normalization}] \quad & \forall y \in \mathcal{Y}: && \textstyle\sum_{i \in \mathcal{I}} M_{y \to i} = 1 \\
[\varepsilon\text{-LabelDP}] \quad & \forall i \in \mathcal{I}, \forall y, y' \in \mathcal{Y} \text{ s.t. } y \neq y': && M_{y' \to i} \leq e^\varepsilon \cdot M_{y \to i} \\
[\text{Unbiasedness}] \quad & \forall y \in \mathcal{Y}: && \textstyle\sum_{i \in \mathcal{I}} M_{y \to i} \cdot \hat{y}_i = y
\end{aligned}
$$

**return** Label randomizer $\mathcal{M}$ mapping $\mathcal{Y}$ to $\hat{\mathcal{Y}}$ given by $\Pr[\mathcal{M}(y) = \hat{y}_i] = M_{y \to i}$.

---

In Figure 2, we illustrate a prior $\mathcal{P}$ over $\mathcal{Y} = \{0, 1, 2\}$, using the example in Table 1, and the corresponding unbiased randomizer returned by ComputeOptUnbiasedRand$_{\varepsilon = 0.5}(\mathcal{P}, \hat{\mathcal{Y}})$ for a certain

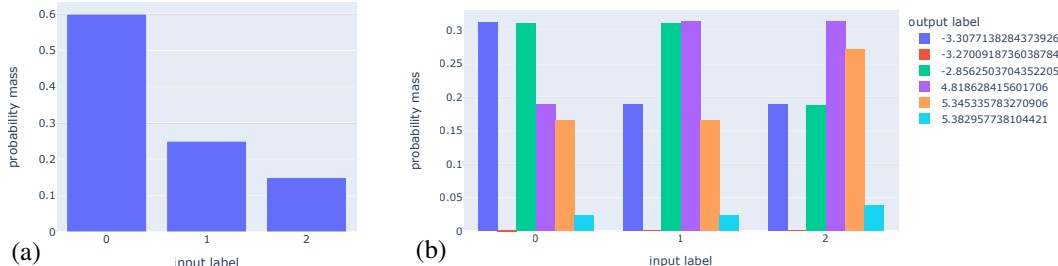

Figure 2: (a) An example prior $\mathcal{P}$ over $\mathcal{Y} = \{0, 1, 2\}$. (b) The optimal unbiased randomizer returned by $\mathsf{ComputeOptUnbiasedRand}_{\varepsilon=0.5}(\mathcal{P}, \hat{\mathcal{Y}})$ using a fine grid for $\hat{\mathcal{Y}}$, indicated by the probability mass function for each input label.

fine-grained choice of $\hat{\mathcal{Y}}$. It is worth noting that this randomizer is quite unlike "randomized response" or any additive noise mechanism! Additionally, the output labels are all outside $[0, 2]$, which is the convex hull of $\mathcal{Y}$. We have observed the same in our experiments in Section 4 as well. One may find it surprising that these "out of domain" noisy labels in fact yield better trained models! This could be attributed to the Bayes optimal predictor for this randomizer being precisely $f_{\mathcal{D}}^*$, since $\mathbb{E}[\mathcal{M}(y)] = y$ for all $y \in \mathcal{Y}$ (Theorem 5).

There are however a couple of challenges to be addressed. First, it is unclear if the optimal unbiased randomizer has only finitely many labels, and even if so, what is the number of distinct output labels. Secondly, one needs to fix the choice of $\hat{\mathcal{Y}}$ before hand, and it is unclear how one can choose it in a way that the LP is feasible and moreover, the solution is close to the optimal unbiased randomizer. We address these challenges as follows.

**Structure of optimal unbiased randomizers.** Addressing the first challenge, we show that there is an optimal unbiased randomizer that has a small number of distinct output labels.

**Theorem 6.** *For any convex loss $g$, all distributions $P$ over $\mathbb{R}$ with finite support, and $\varepsilon > 0$, there exists an $\varepsilon$-DP unbiased randomizer $\mathcal{M}_* \in \inf_{\varepsilon\text{-DP unbiased } \mathcal{M}} \mathcal{G}(\mathcal{M}; \mathcal{P})$ with at most $2|\mathcal{Y}|$ output labels.*

The main ideas in the proof are as follows: we use the convexity of $g$ to show that an optimal $\varepsilon$-DP unbiased randomizer $\mathcal{M}$ must be of a certain form; if not we can use Jensen's inequality to construct another randomizer $\mathcal{M}'$ with that form such that $\mathcal{G}(\mathcal{M}'; P) \leq \mathcal{G}(\mathcal{M}; \mathcal{P})$. In particular, we show that the randomizer must be a *staircase mechanism* as defined in [KOV16], a mechanism that maximally satisfies the DP constraints. This already implies that the number of output labels is at most $2^{|\mathcal{Y}|}$. We use the ordering of the labels, along with similar reductions using convexity of $g$, to show that the optimal $\varepsilon$-DP unbiased randomizer further lies within a special subset of staircase mechanisms with $2|\mathcal{Y}|$ output labels. The detailed proof is provided in Appendix B.

**Choosing $\hat{\mathcal{Y}}$ to ensure feasibility and good coverage.** To address the second challenge, we use a finite set of output labels. We show an upper bound on the excess noisy label loss due to discretization.

We use a heuristic (Algorithm 2) for setting $\hat{\mathcal{Y}}$ to be a grid, in a way that ensures that the LP in Algorithm 1 has a feasible solution while keeping $\hat{\mathcal{Y}}$ small enough to be able to efficiently solve the LP. To compute the endpoints of the grid, we use an unbiased randomizer with a finite and bounded support: the $\varepsilon$-DP *debiased randomized response* ($\varepsilon$-dbRR) on the set of labels, which operates by mapping the inputs to a unique set of values such that under randomized response the randomizer is unbiased (see Appendix F for a definition). We choose the endpoints of our grid to be precisely the minimum and maximum among the possible outputs of $\varepsilon$-dbRR (see Algorithm 2 for details). With those two endpoints defined, we create the grid by evenly spacing output labels along this interval. The number of output labels we generate is as large as possible while maintaining that the LP solver terminates in a reasonable amount of time. We show that having these endpoints suffices to ensure feasibility of the LP in Algorithm 1.

**Proposition 7.** *If $\hat{\mathcal{Y}}$ contains just the smallest and largest values among the outputs of $\varepsilon$-dbRR, the LP in $\mathsf{ComputeOptUnbiasedRand}_{\varepsilon}(P, \hat{\mathcal{Y}})$ is feasible.*

**Algorithm 2** FeasibleOutputSet$_\varepsilon$

---

**Parameters:** Privacy parameter $\varepsilon \geq 0$.
**Input:** Set of input labels $\mathcal{Y}$. Positive integer $n \geq 2$ representing size of output $|\hat{\mathcal{Y}}|$.
**Output:** Set of output values $\hat{\mathcal{Y}}$ that guarantee feasibility of LP in Algorithm 1.
$L \leftarrow \left( (e^\varepsilon + |\mathcal{Y}| - 1) \cdot \min(\mathcal{Y}) - \sum_{y \in \mathcal{Y}} y \right) / (e^\varepsilon - 1)$
$U \leftarrow \left( (e^\varepsilon + |\mathcal{Y}| - 1) \cdot \max(\mathcal{Y}) - \sum_{y \in \mathcal{Y}} y \right) / (e^\varepsilon - 1)$
$\Delta \leftarrow (U - L)/(n - 1)$
**return** $\hat{\mathcal{Y}} = (L, L + \Delta, L + 2\Delta, \ldots, U - \Delta, U)$

---

We defer the proof to Appendix F; the main idea is that these endpoints create an interval that contains the interval one would obtain from the debiased randomized response on the two label set of the minimum label and maximum label. Because the $\hat{\mathcal{Y}}$ we choose has labels less than this minimum and larger than this maximum, all labels in between can be generated by interpolation.

Additionally, we show that using the randomizer optimized on the grid $\hat{\mathcal{Y}}$ gives bounded excess noisy label loss compared to optimizing over a continuous set of labels for Lipschitz loss functions.

**Lemma 8.** *Let $\mathcal{M}$ be the optimal unbiased randomizer with $\hat{\mathcal{Y}} = [L, U]$, and let $\mathcal{M}_\Delta$ be the optimal unbiased randomizer with $\hat{\mathcal{Y}} = \{L, L + \Delta, \ldots, U - \Delta, U\}$. If $g(\hat{y}, y)$ is $K$-Lipschitz in $\hat{y}$, then $\mathcal{G}(\mathcal{M}_\Delta; \mathcal{P}) \leq \mathcal{G}(\mathcal{M}; \mathcal{P}) + K\Delta$.*

In the case of $g(\hat{y}, y) = \frac{1}{2}(\hat{y} - y)^2$, we have $K = (U - L)$. Therefore, as the grid gets finer, the excess noisy label loss of $\mathcal{M}_\Delta$ obtained from 1 over the noisy label loss of $\mathcal{M}$ scales linearly in the discretization parameter $\Delta$. We present the proof in Appendix E.

For large $\varepsilon$ values, we experimentally observe that the optimal unbiased randomizer appears to approach $\varepsilon$-dbRR. For small $\varepsilon$ values, we also experimentally observe that the optimal unbiased randomizer appears to be supported on the labels of the debiased randomized response on the two label set of the minimum label and the maximum label. This justifies our choice of endpoints for grid.

**Splitting budget between prior and label.** So far, we have assumed a known prior $P$ over input labels $\mathcal{Y}$. However, this is typically not the case, and so we use the standard Laplace mechanism to privately estimate the prior. Given $n$ samples drawn from $P$, $\mathcal{M}_\varepsilon^{\mathrm{Lap}}$ constructs a histogram over $\mathcal{Y}$ and adds Laplace noise with scale $2/\varepsilon$ to each entry, followed by clipping (to ensure that entries are non-negative) and normalization. For completeness, we include a formal description of $\mathcal{M}_\varepsilon^{\mathrm{Lap}}$ in Appendix C.

Our randomizer for the unknown prior case, described in Algorithm 3, thus operates by splitting

**Algorithm 3** LabelRandomizer$_{\varepsilon_1, \varepsilon_2}$.

---

**Parameters:** Privacy parameters $\varepsilon_1, \varepsilon_2 \geq 0$.
**Input:** Labels $y_1, \ldots, y_n \in \mathcal{Y}$.
**Output:** $\hat{y}_1, \ldots, \hat{y}_n \in \hat{\mathcal{Y}}$.
$P' \leftarrow \mathcal{M}_{\varepsilon_1}^{\mathrm{Lap}}(y_1, \ldots, y_n)$
$\hat{\mathcal{Y}} \leftarrow$ FeasibleOutputSet$_{\varepsilon_1}(\mathcal{Y})$
$\mathcal{M} \leftarrow$ ComputeOptUnbiasedRand$_{\varepsilon_2}(\mathcal{P}', \hat{\mathcal{Y}})$
**for** $i \in [n]$ **do**
$\quad \hat{y}_i \leftarrow \mathcal{M}(y_i)$
**return** $(\hat{y}_1, \ldots, \hat{y}_n)$

---

the privacy budget into $\varepsilon_1, \varepsilon_2$, using $\mathcal{M}_{\varepsilon_1}^{\mathrm{Lap}}$ to get an approximate prior distribution $P'$, and using the randomizer returned by ComputeOptUnbiasedRand$_{\varepsilon_2}(\mathcal{P}', \hat{\mathcal{Y}})$ to randomize the labels. It follows that the entire algorithm is $(\varepsilon_1 + \varepsilon_2)$-DP.

**Handling continuous $\mathcal{Y}$.** While we focused on the case of finite sets $\mathcal{Y}$, the approach can be extended to the case where $\mathcal{Y}$ is a continuous set, say $\mathcal{Y} = [L, U]$. In this case, we can first choose a finite discrete subset $\widetilde{\mathcal{Y}} \subseteq \mathcal{Y}$ that contains both $L$ and $U$, e.g., $\widetilde{\mathcal{Y}} = \{L, L + \Delta, \ldots, U\}$, and then apply *unbiased randomized rounding* URR$_{\widetilde{\mathcal{Y}}}(\cdot)$ to all labels before applying any mechanism $\mathcal{M}$.

Namely, for any label $y \in [L, U]$ such that $y_1 \leq y \leq y_2$ for $y_1, y_2$ being consecutive elements of $\widetilde{\mathcal{Y}}$, URR$_{\widetilde{\mathcal{Y}}}(y)$ returns $\widetilde{y}$ drawn as $y_1$ with probability $\frac{y_2 - y}{y_2 - y_1}$ and $y_2$ with probability $\frac{y - y_1}{y_2 - y_1}$. This ensures that $\mathbb{E}[\widetilde{y}] = y$ and hence, for any unbiased mechanism $\mathcal{M}$ that acts on inputs in $\widetilde{\mathcal{Y}}$, it holds for all

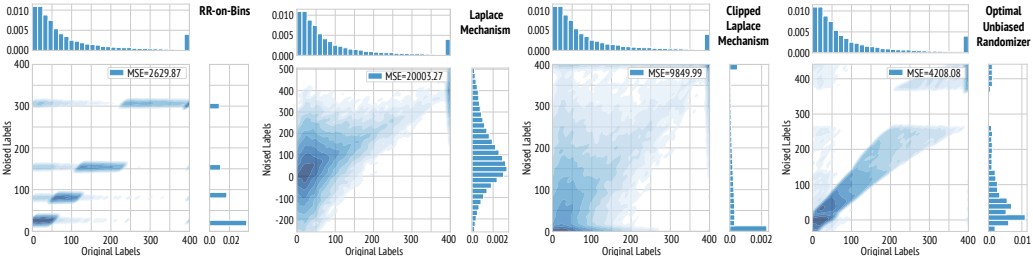

Figure 3: Illustration of various $\varepsilon$-DP label randomizers for $\varepsilon = 4$. The 2D density plot contours are generated in log scale using Gaussian kernel density estimates. The legends show the MSE between the original labels and the $\varepsilon$-DP randomized labels.

$y \in \mathcal{Y}$ that $\mathbb{E}[\mathcal{M}(\mathsf{URR}_{\widetilde{\mathcal{Y}}}(y))] = \mathbb{E}[\mathsf{URR}_{\widetilde{\mathcal{Y}}}(y)] = y$. Furthermore, as the gap between consecutive points of $\widetilde{\mathcal{Y}}$ decreases, the variance introduced by $\mathsf{URR}_{\widetilde{\mathcal{Y}}}$ also decreases.

## 4 Experimental Evaluation

We evaluate our randomizer on three datasets, and compare with the Laplace mechanism [DMNS06], the additive staircase mechanism [GV14], and the RR-on-Bins method [GKK+23]. The Laplace mechanism and the additive staircase mechanism both have a discrete and a continuous variant. Following [GKK+23], we use the continuous variants for real-valued labels (the Criteo Sponsored Search Conversion dataset), and the discrete variants for integer-valued labels (the US Census dataset and the App Ads Conversion Count dataset). Note that in the experiments from [GKK+23], the noised labels from both the Laplace mechanism and the additive staircase mechanism were clipped to be in the valid label value range, as for small $\varepsilon$'s, the magnitude of the noised labels could be orders of magnitudes larger than the original label values, potentially causing numerical instability in model training. However, in our study, we found that with more careful hyperparameter tuning and early stopping (see Appendix G), the learning could be stabilized for sufficiently small values (e.g., $\varepsilon \geq 0.3$) that is practically useful in ML, and in this case, the unclipped (therefore also unbiased) version of both the Laplace and the additive staircase mechanisms outperform their clipped counterpart. For reference, we present the results for both clipped and unclipped variants for those two mechanisms. We note that all of these mechanisms can be implemented in the feature-oblivious label DP setting (Figure 1). More details on model and training configurations are presented in Appendix G.

### 4.1 Criteo Sponsored Search Conversion

The Criteo Sponsored Search Conversion Log Dataset [TY18] is a collection of $15,995,634$ data points derived from a sample of 90-day logs of live traffic from Criteo Predictive Search (CPS). Each example contains information of an user action (e.g., a click on an advertisement) and a subsequent conversion (purchase of the related product) within a 30-day attribution window. We use the setup of feature-oblivious label DP to predict the revenue in € (i.e., the `SalesAmountInEuro` field in the dataset) of a conversion. We apply the following preprocessing steps: filtering out examples with `SalesAmountInEuro` being $-1$, and clipping the labels at the 95th percentile of the value distribution (400€).

In Figure 3, we visualize an example of the various label randomizers. For $\varepsilon = 4$, the optimal RR-on-Bins randomizer maps the input values to one of four distinct values. On the other hand, for the optimal unbiased randomizer, the joint distribution of the sensitive labels and randomized labels maintains an overall concentration along the diagonal. The joint distribution for the Laplace mechanism is quite spread out.

In Figure 4, we compare the noisy label loss on the training set and the mean squared error on the test set for all the randomizers considered. For each randomizer, the label randomization and training was run with 10 independent random seeds, and the plots show the average and standard deviation bars. For RR-on-Bins and the optimal unbiased randomizer, we use $\varepsilon_1 = 0.017$ for privately estimating the prior using $\mathcal{M}_{\varepsilon_1}^{\mathrm{Lap}}$, and $\varepsilon_2 = \varepsilon - \varepsilon_1$ for randomizing the label, following the guidance in Appendix C.

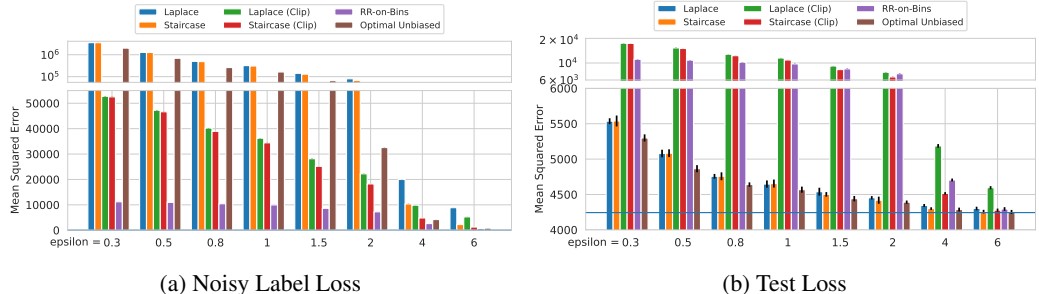

(a) Noisy Label Loss

(b) Test Loss

Figure 4: Comparison of different label DP randomizers on the Criteo Search dataset: (a) shows the mean squared error between the original labels and the privatized labels on the training set from each randomizer; (b) shows the mean squared error between the model predictions and the groundtruth labels on the test set. The solid line indicates the non-private baseline.

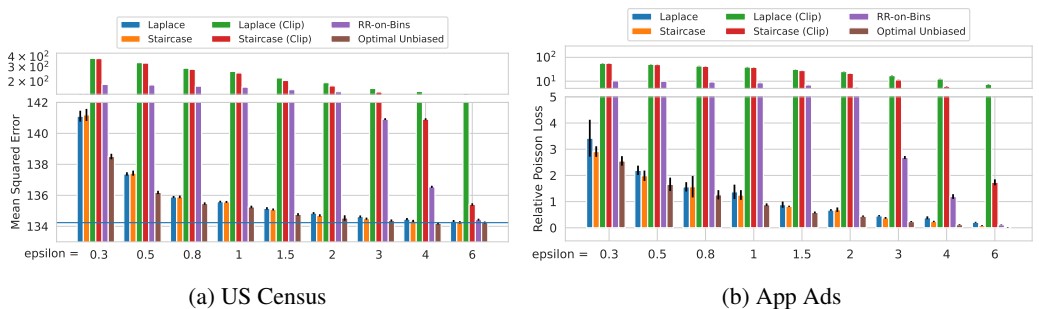

(a) US Census

(b) App Ads

Figure 5: Comparison of different label DP randomizers: (a) shows the mean squared error of the prediction on the US Census test set. The solid line indicates the non-private baseline. (b) shows the relative Poisson loss (i.e., $(L - L^\star)/L^\star$) with respect to the non-private baseline Poisson loss $L^\star$.

We find that the optimal unbiased randomizer (Algorithm 3) achieves the smallest test mean squared error across a wide range of $\varepsilon$ values. It is interesting to note that the unbiased randomizers (Laplace and additive staircase mechanisms and the optimal unbiased randomizer) vastly outperform the biased randomizers (clipped versions of Laplace and additive staircase mechanisms as well as the RR-on-Bins) in terms of test loss, even though the noisy label loss is an order of magnitude larger for the unbiased randomizers.

## 4.2 US Census

The 1940 US Census dataset[3] is widely used in the evaluation of data analysis with DP [WDZ+19, CJG21, GKM+21]. This digitally released dataset in 2012 contains $131,903,909$ examples. We follow the task studied in [GKK+23] and evaluate label DP algorithms by learning to predict the number of weeks each respondent worked during the previous year (the WKSWORK1 field).

In Figure 5a we compare the mean squared error on the test set for all the randomizers considered. For each randomizer, the label randomization and training was run with 10 independent random seeds, and the plots show the average and standard deviation bars. For RR-on-Bins and the optimal unbiased randomizer, we use $\varepsilon_1 = 0.002$ for privately estimating the prior using $\mathcal{M}_{\varepsilon_1}^{\mathrm{Lap}}$, and $\varepsilon_2 = \varepsilon - \varepsilon_1$ for randomizing the label, following the guidance in Appendix C. We find that the optimal unbiased randomizer achieves the smallest test mean squared error across a wide range of $\varepsilon$ values.

---

[3]https://www.archives.gov/research/census/1940

### 4.3 App Ads Conversion Count

We also evaluate on a conversion count prediction dataset collected from a commercial mobile app store. Each example in this dataset corresponds to an ad click and the task is to predict the number of post-click conversion events in the app after a user installs the app within a certain time window.

In Figure 5b we compare the relative Poisson loss on the test set for all the randomizers considered. For each randomizer, the label randomization and training was run with 6 independent random seeds. For the optimal unbiased randomizer, Laplace and additive staircase mechanisms, at low $\varepsilon$'s we experience blowup in the training loss during training. The plot show the average and standard deviation bars, at the time of lowest training loss, right before blowup. We see that for all $\varepsilon$'s, all of the unbiased randomizers vastly outperform the others, including RR-on-Bins. Among the unbiased randomizers (optimal unbiased randomizer, Laplace and additive staircase mechanisms), the optimal unbiased randomizer performs the best at all $\varepsilon$'s. One caveat is that for the smaller $\varepsilon$'s, one can see the standard error increase. We hypothesize this is not an inherent feature of the randomizers but due to the blow up of the model training, reducing the training length and adding additional variance to the error, a behavior that seems specific to the AppAds dataset.

## 5 Related Work

DP learning has been the subject of considerable research spanning different settings that include empirical risk minimization [CMS11], PAC learning [BLM20], training neural networks [ACG$^+$16], online learning [JKT12], and regression [Wan18]. For the label DP setting, [CH11, BNS16] studied the sample complexity of classification, while [WX19] studied sparse linear regression in the local DP model [EGS03, KLN$^+$11]. For training deep neural networks with label DP, [GGK$^+$21] studied the classification setting and gave a multi-stage training procedure where the priors on the labels in the previous stage are used to define an LP that is optimized to find the randomization mechanism for the next stage. For the classification loss, they characterized the optimal solution as the so-called RRTop-$k$. By contrast, the work of [GKK$^+$23] showed that for regression objectives the optimal solution to the LP is an RR-on-Bins solution. A crucial insight in our work is that the addition of an unbiasedness constraint to the LP leads to solutions that (i) are not RR-on-Bins solutions, (ii) can have substantially higher variance than RR-on-Bins, and (iii) nevertheless have a much lower train and test error due to the reduction in bias.

Kairouz et al. [KOV16] defined a family of staircase mechanisms and showed they are optimal among local DP algorithms for minimizing an objective function given a prior. We extend the notion of staircase mechanisms to include real-valued labels that affect the optimization function. The structural properties that we prove (Theorem 6) utilize the ordering of the labels and the unbiasedness condition, which we show is critical for training neural networks for regression with label DP and high accuracy. Moreover, their work shows the presence of staircase mechanisms in the context of optimizing mutual information and KL-divergence objectives, whereas we show their presence in the context of regression.

A two-party learning setting with one features party and one labels party was recently studied in the work of [LSY$^+$21]. They focus on the interactive setting, which is arguably less practical than the (non-interactive) feature-oblivious setting studied in [GKK$^+$23] that we also consider.

We also note that the label DP mechanism of [MEMP$^+$21], which builds on the PATE framework [PAE$^+$17, PSM$^+$18], and the works of [EMSV22, TNM$^+$22], which leverage unsupervised and semi-supervised learning algorithms, cannot be implemented in the feature-oblivious setting. The DP-SGD algorithm of [ACG$^+$16], which protects both the features and the label of each training example and which has been applied to different domains including computer vision [DBH$^+$22] and language models [YNB$^+$22] also cannot implemented in the feature-oblivious label DP setting. The same is also true for label DP algorithms for logistic regression [GR21] that are based on linear queries (where the features are assumed to be known and non-sensitive, and the linearity is with respect to the labels).

Over the past decade, there has been a significant body of research on DP ML (e.g., [CMS11, ZZX$^+$12, SCS13]). In particular, DP regression has been the focus of several prior papers including [ZZX$^+$12, KST12, Wan18, SÁZL18, AMS$^+$22].

The label DP setting has also been studied in several papers including [CH11, BNS16, WX19, GGK+21, MEMP+21, YSMN21, EMSV22, BMS22, GKK+23]. The Randomized Response (RR) mechanism [War65], a basic form of label DP, was introduced several decades ago and is widely studied/used.

# 6   Conclusion

In this work, we show that using unbiased $\varepsilon$-DP label randomizers lead to better trained models, and in particular, choosing the label randomizer that minimizes the noisy label squared loss seems to perform the best in terms of test performance, by empirically demonstrating this on three datasets. We also provide theoretical results shedding light on the structure of these randomizers, as well as why they might lead to better trained models.

**Discussion.**   While we focused entirely on $\varepsilon$-DP ("pure-DP"), for our specific approach, it does not seem that relaxing to $(\varepsilon, \delta)$-DP ("approximate-DP") will be beneficial. For the first stage of estimating the histogram privately, one could potentially use an approximate-DP mechanism, but we believe that is unlikely to change the prior significantly. For the second stage of randomizing labels, it is known that approximate-DP may not be helpful in the local model [BNS18].

In this work, we use hyperparameter tuning to choose the best architecture, which in general has additional privacy costs, and how to tune hyperparameters privately and efficiently is an active research topic [PS22, SSS23]. Consequently, it is common in the private ML literature to separate the question of private hyperparameter tuning and private training, and focus on comparing the privacy-utility trade-off under optimal hyperparameters, e.g., [MEMP+21, HLY+23, KCS+22, DBH+22]. We also follow this convention.

**Limitations and future work.**   While LPs are somewhat efficient, it would be desirable to have a more efficient algorithm for computing the optimal unbiased randomizer. The RR-on-Bins randomizer introduced in [GKK+23] has an important advantage that one does not need to construct the set of potential output labels $\hat{\mathcal{Y}}$ before hand, and in fact, there is a simple dynamic programming algorithm to compute the optimal randomizer along with the output labels. Moreover, the RR-on-Bins family of randomizers has a clean structure. Understanding more structural properties of the optimal unbiased randomizers, and designing better algorithms for computing them, would be an interesting future direction to pursue.

To further improve the benefit of this optimal prior-based unbiased label randomizer, it might be possible to first partition the input examples into groups of "related examples", purely using the input features and compute the optimal unbiased randomizer for each group, by privately estimating the prior over labels for each group. This could lead to better test performance, as the randomizer can adapt to the different priors across these groups, unlike the "static" label randomizers such as Laplace and additive staircase mechanisms. For example, such an approach was used in the setting of image classification using self-supervised learning in [GGK+21].

As shown in (2), there is a bias-variance trade-off when it comes to choosing a label randomizer. While RR-on-Bins minimizes the noisy label loss (corresponding to the variance), the method in our work minimizes the bias (in fact, making it zero), at the cost of greatly increasing the variance. However, it might be possible in some settings that allowing a small amount of bias might greatly reduce the variance, thereby improving performance. This might be especially relevant when $\varepsilon$ is very small. This could be done for example by adding a regularizer to reduce the bias, without enforcing hard unbiased constraints, as currently done in Algorithm 1. We leave this investigation to future work.

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

# A  Bayes Optimal Predictor Using Noisy Labels

We prove Theorem 5, restated below for convenience.

**Theorem 5.** *Suppose $\ell$ is a loss function such that for all distributions $\mathcal{P}$ over $\mathcal{Y}$, the minimizer $\hat{y}_* := \min_{\hat{y} \in \mathbb{R}} \mathbb{E}_{y \sim \mathcal{P}} \ell(\hat{y}, y)$ exists and is given as $\hat{y}_* = \mathbb{E}_{y \sim \mathcal{P}}[y]$. Then the following are equivalent:*

▷ $\mathbb{E}[\mathcal{M}(y)] = y$ *holds for all $y \in \mathcal{Y}$,*
▷ *For all distributions $\mathcal{D}$ over $\mathcal{X} \times \mathcal{Y}$, it holds that $f_{\mathcal{D}}^* = f_{\mathcal{D}_{\mathcal{M}}}^*$.*

*Proof.* If $\mathcal{M}$ is unbiased, then $\mathbb{E}_{\hat{y} \sim \mathcal{M}(y)}[\hat{y}] = \sum_{\hat{y} \in \hat{\mathcal{Y}}} \hat{y} \cdot \Pr[\mathcal{M}(y) = \hat{y}] = y$ for all input labels $y \in \mathcal{Y}$. We have the following equalities (we restrict ourselves to the case of finite $\mathcal{Y}$ and $\hat{\mathcal{Y}}$, but the proof generalizes easily to measurable sets $\mathcal{Y}$ and $\hat{\mathcal{Y}}$):

$$
\begin{aligned}
\mathop{\mathbb{E}}_{\substack{\hat{y} \sim \mathcal{M}(y) \\ (x,y) \sim \mathcal{D}}}[\hat{y} \mid x = x_0] 
&= \sum_{y \in \mathcal{Y}, \hat{y} \in \hat{\mathcal{Y}}} \hat{y} \cdot \Pr[\hat{y} \mid y, x = x_0] \cdot \Pr[y \mid x = x_0] \\
&= \sum_{y \in \mathcal{Y}, \hat{y} \in \hat{\mathcal{Y}}} \hat{y} \cdot \Pr[\hat{y} \mid y] \cdot \Pr[y \mid x = x_0] \quad (\because \text{feature obliviousness}) \\
&= \sum_{y \in \mathcal{Y}} \Pr[y \mid x = x_0] \cdot \left( \sum_{\hat{y} \in \hat{\mathcal{Y}}} \hat{y} \cdot \Pr[\hat{y} \mid y] \right) \\
&= \sum_{y \in \mathcal{Y}} \Pr[y \mid x = x_0] \cdot y \quad (\because \text{unbiasedness}) \\
&= \mathop{\mathbb{E}}_{y \sim D(x_0)}[y \mid x = x_0],
\end{aligned}
$$

showing that the Bayes optimal predictors are equal at all values $x_0$.

Conversely, suppose we have a $y' \in \mathcal{Y}$ such that $\mathbb{E}_{\hat{y} \sim \mathcal{M}(y')}[\hat{y}] \neq y'$. Then take any distribution $D$ for which there is some $x_0$ with $\Pr[y = y' \mid x = x_0] = 1$. Then clearly $\mathbb{E}_{y \sim D(x_0)}[y \mid x = x_0] = y'$. However, $\mathbb{E}_{\hat{y} \sim \mathcal{M}(y), y \sim D(x_0)}[\hat{y} \mid x = x_0] = \mathbb{E}_{\hat{y} \sim \mathcal{M}(y')}[\hat{y}] \neq y'$. Hence the Bayes optimal predictors differ at $x_0$. $\qquad\square$

# B  Structural Properties of Optimal Unbiased Randomizers

We prove Theorem 6, restated below for convenience. We use the following assumption about the label loss $g$, which in particular, is satisfied by $\ell_{\mathrm{sq}}$ that we primarily use.

**Assumption 9.** *The loss $g : \mathbb{R} \times \mathbb{R} \to \mathbb{R}$ is such that for all $y$, it holds that $g(\hat{y}, y)$ is convex in $\hat{y}$, and minimized at $\hat{y} = y$ for all $y$.*

**Theorem 6.** *For any convex loss $g$, all distributions $P$ over $\mathbb{R}$ with finite support, and $\varepsilon > 0$, there exists an $\varepsilon$-DP unbiased randomizer $\mathcal{M}_* \in \inf_{\varepsilon\text{-DP unbiased } \mathcal{M}} \mathcal{G}(\mathcal{M}; \mathcal{P})$ with at most $2|\mathcal{Y}|$ output labels.*

We recall the definition of a signature matrix from [GKK$^+$23]. Consider any feasible solution $M := (M_{y \to i})_{y \in \mathcal{Y}, i \in \mathcal{I}}$ of the LP in Algorithm 1. Note that for reasons that will be clear shortly, we do not require the $\hat{y}_i$ values to be distinct, and we will assume without loss of generality that $\hat{y}_i \leq \hat{y}_j$ for all $i < j \in \mathcal{I}$. Let $p_i^{\min} = \min_y M_{y \to i}$ and let $p_i^{\max} = \max_y M_{y \to i}$. Note that if $M$ encodes an $\varepsilon$-DP randomizer, it holds that $p_i^{\max} \leq e^{\varepsilon} \cdot p_i^{\min}$.

**Definition 10** (Signature matrix). For any $\varepsilon$-DP randomizer encoded by $M$, the corresponding *signature* matrix entry $S_M(y, i)$ for all $y \in \mathcal{Y}$ and $i \in \mathcal{I}$ is defined as

$$
S_M(y, i) = \begin{cases}
0 & \text{if } M_{y \to i} = 0 \\
\mathsf{U} & \text{if } M_{y \to i} = p_i^{\max} = e^{\varepsilon} \cdot p_i^{\min} \\
\mathsf{L} & \text{if } M_{y \to i} = p_i^{\min} = e^{-\varepsilon} \cdot p_i^{\max} \\
\mathsf{S} & \text{otherwise.}
\end{cases}
$$

We show that for any $\varepsilon$-DP unbiased randomizer $\mathcal{M}^{(0)}$, there exists another $\varepsilon$-DP unbiased randomizer $\mathcal{M}$, which satisfies certain nice properties, while not increasing the noisy label loss.

**Claim 11.** *Suppose $|\mathcal{Y}| \geq 2$. For any $\varepsilon$-DP unbiased randomizer $\mathcal{M}^{(0)}$ mapping to a finite set of output labels, there exists an $\varepsilon$-DP unbiased randomizer $\mathcal{M}$ with output label sequence $\hat{\mathcal{Y}} = (\hat{y}_i)_{i \in \mathcal{I}}$, encoded by $(M_{y \rightarrow i})_{y,i}$ such that the following hold:*

> $\triangleright$ $\mathcal{G}(\mathcal{M}^0; P) \geq \mathcal{G}(\mathcal{M}; P)$.
> $\triangleright$ *Each column of $S_M$ consists of only $\mathsf{U}$'s and $\mathsf{L}$'s, with at least one $\mathsf{U}$ and one $\mathsf{L}$.*
> $\triangleright$ *With row indices in $\mathcal{Y}$ sorted in increasing order from top to bottom, every non-zero column of $S_M$ matches the regular expression $\mathsf{L}^*\mathsf{U}^+\mathsf{L}^*$. For each $i \in \mathcal{I}$, let $\Phi_1(i)$ and $\Phi_2(i)$ be the smallest and the largest $y$ respectively for which $S_M(y, i) = \mathsf{U}$.*
> $\triangleright$ *For all $i < i'$ it holds that $\Phi_1(i) \leq \Phi_1(i')$ and $\Phi_2(i) \leq \Phi_2(i')$, with at least one of the inequalities being strict.*

Before we prove Claim 11, we prove Theorem 6 using it.

*Proof of Theorem 6.* If $|\mathcal{Y}| = 1$, then it is immediate that the optimal unbiased randomizer simply returns the unique value in $\mathcal{Y}$ with probability 1. If $|\mathcal{Y}| \geq 2$, then for any optimal unbiased randomizer $\mathcal{M}^{(0)}$, we can apply Claim 11 to get another optimal unbiased randomizer $\mathcal{M}$ that satisfies the stated conditions. Let $\mathcal{I}_{\neq 0} \subseteq \mathcal{I}$ be the subsequence of all outputs for which $M_{y \rightarrow i} \neq 0$ (for all $y$). The sequence $(\Phi_1(i), \Phi_2(i))$ ordered in increasing order of $i \in \mathcal{I}_{\neq 0}$ is strictly increasing in the partial order $\mathcal{Y} \times \mathcal{Y}$, that is, at least one of $\Phi_1(i)$ or $\Phi_2(i)$ strictly increases from one $i$ to next. It is easy to see that this can happen at most $2|\mathcal{Y}|$ times, thereby concluding that $|\mathcal{I}_{\neq 0}| \leq 2|\mathcal{Y}|$. $\qquad \square$

To prove Claim 11, we perform several transformations to the given $\varepsilon$-DP randomizer $\mathcal{M}^{(0)}$, which is say encoded by $M^{(0)} = (M^{(0)}_{y \rightarrow i})_{y,i}$ for some $\hat{\mathcal{Y}} = (\hat{y}_i)_{i \in \mathcal{I}}$, such that the randomizer obtained in each step satisfies $\mathbb{E}\,\mathcal{M}^{(r)}(y) = \mathbb{E}\,\mathcal{M}^{(r+1)}(y)$ for all $y \in \mathcal{Y}$ and $\mathcal{G}(\mathcal{M}^{(r)}; P) \geq \mathcal{G}(\mathcal{M}^{(r+1)}; P)$.

**Combining columns with identical signatures.** For convenience let $k := |\mathcal{Y}|$. Let $b_j$ be the $k$-dimensional binary vector corresponding to the binary representation of $j$ for $j \leq 2^k - 1$. A matrix $S^{(k)} \in \{1, e^\varepsilon\}^{k \times (2^k - 2)}$ is called a *Staircase Pattern Matrix* [KOV16] if the $j$th column of $S^{(k)}$ is $S^{(k)}_j = (e^\varepsilon - 1)b_j + 1$, for $j \in \{1, \ldots, 2^k - 2\}$. For example,

$$S^{(3)} = \begin{bmatrix} 1 & 1 & 1 & e^\varepsilon & e^\varepsilon & e^\varepsilon \\ 1 & e^\varepsilon & e^\varepsilon & 1 & 1 & e^\varepsilon \\ e^\varepsilon & 1 & e^\varepsilon & 1 & e^\varepsilon & 1 \end{bmatrix}$$

Since $\mathcal{M}^{(0)}$ is $\varepsilon$-DP, we have that each column of $M^{(0)}$, namely $(M^{(0)}_{y \rightarrow i})_{y \in \mathcal{Y}}$ is a conic combination of the columns of $S^{(k)}$.

**Claim 12.** *Let $\mathcal{M}^{(0)}$ be an $\varepsilon$-DP randomizer, mapping $\mathcal{Y}$ to a finite subset $\hat{\mathcal{Y}}^{(0)}$ of $\mathbb{R}$. Then there exists an $\varepsilon$-DP randomizer $\mathcal{M}^{(1)}$ mapping $\mathcal{Y}$ to $\hat{\mathcal{Y}}^{(1)} \subseteq \mathbb{R}$ encoded by $(M_{y \rightarrow \hat{y}})_{y \in \mathcal{Y}, \hat{y} \in \hat{\mathcal{Y}}^{(1)}}$ such that:*

> $\triangleright$ $\mathcal{G}(\mathcal{M}^{(1)}; \mathcal{P}) \leq \mathcal{G}(\mathcal{M}^{(0)}; \mathcal{P})$ *for any convex loss $g$.*
> $\triangleright$ $\mathbb{E}[\mathcal{M}^{(1)}(y)|y] = \mathbb{E}[\mathcal{M}^{(0)}(y)|y]$.
> $\triangleright$ $|\hat{\mathcal{Y}}| \leq 2^{|\mathcal{Y}|}$.
> $\triangleright$ *The vector $(M_{y \rightarrow \hat{y}})_{y \in \mathcal{Y}}$ is a positive multiple of a staircase column, with the staircase column being unique for each $\hat{y} \in \hat{\mathcal{Y}}^{(1)}$.*

The proof idea of combining columns of staircase type to potentially decrease loss was shown in [KOV16].

*Proof.* By $\varepsilon$-DP constraint, the columns of $M^{(0)}$ lie in the cone made by convex combinations of the rays $\{c, c \cdot e^\varepsilon\}^{|\mathcal{Y}|}$ where $c \geq 0$. Because positive multiples of the staircase columns are specifically these rays, except the ray $\{c\}^{|\mathcal{Y}|}$ which is already in the convex hull, this column of $\mathcal{M}^{(0)}$ is a convex combination of positive multiples of the staircase columns.

Let $v_{\hat{y}}$ be the columns of $M^{(0)}$ indexed by $\hat{\mathcal{Y}}^{(0)}$, which by the above can be written as:

$$v_{\hat{y}} = \sum_{j=1}^{2^k-2} c_{\hat{y}}^j \cdot S_j^{(k)}.$$

Define for each $j$:

$$C^j = \sum_{\hat{y} \in \hat{\mathcal{Y}}^0} c_{\hat{y}}^j \qquad \text{and} \qquad \hat{y}_j = \frac{\sum_{\hat{y} \in \hat{\mathcal{Y}}^0} \hat{y} \cdot c_{\hat{y}}^j}{C^j}.$$

Define $M^{(1)}$ as being supported only at $\hat{y}_j$ with the vector of probabilities being $C^j \cdot S_j^{(k)}$. We now prove $M^{(1)}$ has all the desired properties.

▷ (Defines a Mechanism):

$$\sum_{j=1}^{2^k-2} C^j \cdot S_j^{(k)} = \sum_{j=1}^{2^k-2} \left( \sum_{\hat{y} \in \hat{\mathcal{Y}}^0} c_{\hat{y}}^j \right) \cdot S_j^{(k)} = \sum_{\hat{y} \in \hat{\mathcal{Y}}^0} v_{\hat{y}} = \vec{1}.$$

▷ (Does not increase loss): This follows by Jensen's inequality and the linearity of expectation:

$$\mathcal{G}(M^{(0)}; \mathcal{P}) = \sum_{j=1}^{2^k-2} \sum_{y \in \mathcal{Y}} (S_j^{(k)})_y \cdot \left( \sum_{\hat{y}} g(\hat{y}, y) \cdot c_{\hat{y}}^j \right) \geq \sum_{j=1}^{2^k-2} \sum_{y \in \mathcal{Y}} (S_j^{(k)})_y \cdot \left( g(\hat{y}_j, y) \cdot C^j \right) = \mathcal{G}(M^{(1)}; \mathcal{P}).$$

▷ (Expectation does not change): Because $M^{(0)}$ is unbiased:

$$\sum_{\hat{y} \in \hat{\mathcal{Y}}^0} \hat{y} \cdot v_{\hat{y}} = (y)_{y \in \hat{Y}}.$$

We have

$$\sum_{j=1}^{2^k-2} \hat{y}_j \cdot C^j \cdot S_j^{(k)} = (y)_{y \in \hat{Y}},$$

showing that $M^{(1)}$ is unbiased.

▷ (Bounded by $2^{|\mathcal{Y}|}$ and multiple of unique staircase columns): This is clear by construction.  □

While the finite case is easier to read, the above proof can be done also when the range of $M^{(0)}$ is $\mathbb{R}$ and $v_{\hat{y}}$ is a vector of absolutely continuous probability distributions. The only change is that sums over $\hat{y} \in \hat{\mathcal{Y}}^{(0)}$ become integrals. The projections of $v_{\hat{y}}$ onto the staircase columns are continuous maps. Lastly, because a continuous function that is bounded by a function in $L^1$ is itself in $L^1$, all the integrals become well-defined and finite. Therefore, our theorem regarding the structure of the optimal unbiased $\varepsilon$-DP mechanism does not depend on a finiteness condition.

**Each row matches the pattern $\mathsf{L}^*\mathsf{U}^+\mathsf{L}^*$.** We now use the ordering of the input and output labels as real numbers to make further deductions.

**Claim 13.** *Suppose $|\mathcal{Y}| \geq 2$. For any $\varepsilon$-DP randomizer $\mathcal{M}^{(1)}$ obtained via Claim 12, there exists an $\varepsilon$-DP randomizer $\mathcal{M}^{(2)}$ mapping $\mathcal{Y}$ to $\hat{\mathcal{Y}} \subseteq \mathbb{R}$ encoded by $M := (M_{y \to \hat{y}})_{y \in \mathcal{Y}, \hat{y} \in \hat{y}}$ such that:*

▷ *$\mathcal{G}(\mathcal{M}^{(2)}; \mathcal{P}) \leq \mathcal{G}(\mathcal{M}^{(1)}; \mathcal{P})$ for any convex loss g,*
▷ *$\mathbb{E}[\mathcal{M}^{(2)}(y)|y] = \mathbb{E}[\mathcal{M}^{(1)}(y)|y],$*
▷ *Every row of $S_M$ matches the regular expression $\mathsf{L}^*\mathsf{U}^+\mathsf{L}^*$.*

*Proof.* The only way a row of $S_M$ does not match the regular expression $\mathsf{L}^*\mathsf{U}^+\mathsf{L}^*$ is if (i) there is no $\mathsf{U}$ present, or (ii) if there is an $\mathsf{L}$ between an two $\mathsf{U}$'s. It is easy to see that (i) is not possible because $\sum_{\hat{y}} M_{y \to i} = 1$ for all $y \in \mathcal{Y}$, and if one row has no $\mathsf{U}$'s, this would imply no row has any $\mathsf{U}$'s, which is not possible by the uniqueness of the columns if $|\mathcal{Y}| \geq 2$, and we know such an $\mathcal{M}$ exists.

If (ii) is true, then we can construct $\mathcal{M}^{(2)}$ as follows. For any $y$ and $\hat{y}_1 < \hat{y}_2 < \hat{y}_3$ such that $S_{M^{(1)}}(y, \hat{y}_1) = S_{M^{(1)}}(y, \hat{y}_3) = \mathsf{U}$ and $S_{M^{(1)}}(y, \hat{y}_2) = \mathsf{L}$, consider a perturbation (for a small enough $\eta > 0$), where we set

$$
\begin{aligned}
M^{(2)}_{y \to \hat{y}_1} &= M^{(1)}_{y \to \hat{y}_1} - (y_3 - y_2)\eta, \\
M^{(2)}_{y \to \hat{y}_2} &= M^{(1)}_{y \to \hat{y}_2} + (y_3 - y_1)\eta, \\
M^{(2)}_{y \to \hat{y}_3} &= M^{(1)}_{y \to \hat{y}_3} - (y_2 - y_1)\eta.
\end{aligned}
$$

This choice ensures that $M^{(2)}$ remains an unbiased randomizer. By Jensen's inequality, due to convexity of $g$, it holds that $\mathcal{G}(\mathcal{M}^{(2)}; P) < \mathcal{G}(\mathcal{M}^{(1)}; P)$. We can keep repeating these steps, in addition to Claim 12 if necessary, till we arrive at $\mathcal{M}^{(2)}$ such that each row of $S_{M^{(2)}}$ matches the regular expression $\mathsf{L}^*\mathsf{U}^+\mathsf{L}^*$. $\qquad\square$

**Set of U's in any row cannot be a subset of the set of U's in another, and are "moving forward".**

**Claim 14.** *For an $\varepsilon$-DP unbiased randomizer $\mathcal{M}^{(2)}$ obtained via Claim 13, let $T(y) \subseteq \hat{\mathcal{Y}}$ be defined as $T(y) := \{\hat{y} : S_M(y, \hat{y}) = \mathsf{U}\}$. Then, for all $y \neq y'$, it holds that $T(y) \not\subset T(y')$, and moreover if $y < y'$, then $\min T(y) < \min T(y')$ and $\max T(y) < \max T(y')$.*

*Proof.* If $T(y) \subset T(y')$, then it holds that $M_{y \to \hat{y}} \leq M_{y' \to \hat{y}}$ with the inequality being strict for some $\hat{y}$. Thus, we have $\sum_{\hat{y}} M_{y \to \hat{y}} < \sum_{\hat{y}} M_{y' \to \hat{y}} = 1$, which is a contradiction.

Given that both $T(y)$ and $T(y')$ are contiguous subsets of $\hat{\mathcal{Y}}$, it follows that one of following holds:

▷ $\min T(y) < \min T(y')$ and $\max T(y) < \max T(y')$, or
▷ $\min T(y) > \min T(y')$ and $\max T(y) > \max T(y')$.

However, if $y < y'$, the latter is not possible because in that case, we would have $y' = \mathbb{E}_{\hat{y} \sim \mathcal{M}^{(2)}}(y') < \mathbb{E}_{\hat{y} \sim \mathcal{M}^{(2)}}(y) = y$, which is a contradiction. $\qquad\square$

**Putting together the claims.** Finally, we put together Claims 12 to 14 to prove Claim 11.

*Proof of Claim 11.* Given any $\varepsilon$-DP unbiased randomizer $\mathcal{M}$, we apply Claims 12 and 13 to obtain the $\varepsilon$-DP unbiased randomizer $\mathcal{M}^{(2)}$, which satisfies the conditions in Claim 14. Consider the signature matrix $S_M$ corresponding to $\mathcal{M}^{(2)}$, with the row indices sorted in increasing order of $\mathcal{Y}$.

If a column has a signature with an $\mathsf{L}$ between two $\mathsf{U}$'s, this would contradict Claim 14, since the $\mathsf{U}$'s only "move forward".

Additionally, for the signatures of column $\hat{y}_1 < \hat{y}_2$, the first $\mathsf{U}$ of column $\hat{y}_2$ cannot come before the first $\mathsf{U}$ of column $\hat{y}_1$, as this would also contradict the claim of the $\mathsf{U}$'s moving forward. Similarly, the last $\mathsf{U}$ of column $\hat{y}_1$ cannot come after the last $\mathsf{U}$ of column $\hat{y}_2$. Finally, the columns must be unique, since each column is a unique staircase matrix vector. Thus, each column of $S_M$ matches the regular expression $\mathsf{L}^*\mathsf{U}^+\mathsf{L}^*$. $\qquad\square$

## C    Optimal Unbiased Randomizers with Approximate Prior

We elaborate on the approach of splitting privacy budget for estimating the prior and for generating noisy labels as described in Section 3.

To privately estimate the prior, we use the Laplace mechanism. The Laplace distribution with scale parameter $b$, denoted by $\mathrm{Lap}(b)$, is the distribution supported on $\mathbb{R}$ whose probability density function is $\frac{1}{2b}\exp(-|x|/b)$. The Laplace mechanism is presented in Algorithm 4.

It is well-known (e.g., [DMNS06]) that this mechanism satisfies $\varepsilon$-DP. It is also well-known that this yields an approximation of the true distribution up to small total variation distance (e.g., [DHS15]).

In [GKK$^+$23], it is argued that the best choice of $(\varepsilon_1, \varepsilon_2)$ that minimizes the excess noisy label loss achieved by randomizer computed in Algorithm 3 over the optimal unbiased randomizer computed using the true prior and privacy parameter $\varepsilon$, is given as $\varepsilon_1 = \sqrt{k/n}$. (We follow the same budget splitting method in our experiments.)

---

**Algorithm 4** Laplace Mechanism for Estimating Probability Distribution $\mathcal{M}_\varepsilon^{\mathrm{Lap}}$.

---

**Parameters:** Privacy parameter $\varepsilon \geq 0$.
**Input:** Labels $y_1, \ldots, y_n \in \mathcal{Y}$.
**Output:** A probability distribution $P'$ over $\mathcal{Y}$.

**for** $y \in \mathcal{Y}$ **do**
    $h_y \leftarrow$ number of $i$ such that $y_i = y$
    $h'_y \leftarrow \max\{h_y + \mathrm{Lap}(2/\varepsilon), 0\}$
**return** Distribution $P'$ over $\mathcal{Y}$ such that $p'_y = \frac{h'_y}{\sum_{y \in \mathcal{Y}} h'_y}$

---

# D  Excess Loss of SGD with Biased Gradient Oracles

Consider a *convex* objective $\mathcal{L}(\cdot)$ over $\mathbb{R}^D$, for which we have a gradient oracle, that given $w$, returns a stochastic estimate $g(w)$ of $\nabla \mathcal{L}(w)$. We say that the gradient oracle has bias $\alpha$ and variance $\sigma^2$ if $g(w) = \nabla\mathcal{L}(w) + \zeta(w)$ such that $\|\mathbb{E}\,\zeta(w)\| \leq \alpha$ and $\mathbb{E}\|\zeta(w) - \mathbb{E}\,\zeta(w)\|^2 \leq \sigma^2$ holds for all $w$. When optimizing over a convex set $\mathcal{K} \subseteq \mathbb{R}^D$, projected GD with step size $\eta$ is defined as iteratively performing $w_{t+1} \leftarrow \Pi_\mathcal{K}(w_t - \eta g(w_t))$, where $\Pi_\mathcal{K}(\cdot)$ is the projection onto $\mathcal{K}$. We recall the following guarantee on the expected excess loss using a standard analysis (see [Haz22]).

**Lemma 15.** *For a $L$-Lipschitz loss function, and a gradient oracle with bias $\alpha$ and variance $\sigma^2$, projected GD over a set $\mathcal{K}$ with diameter $R$, with step size $\eta = \frac{R}{\sqrt{((L+\alpha)^2 + \sigma^2)T}}$ achieves*

$$\mathbb{E}\left[\mathcal{L}\left(\tfrac{1}{T}\sum_{i=1}^T w_i\right)\right] - \mathcal{L}(w^*) \leq \tfrac{RL}{\sqrt{T}} \cdot \sqrt{1 + \tfrac{2\alpha}{L} + \tfrac{\alpha^2 + \sigma^2}{L^2}} + \alpha R.$$

*In particular, if $\alpha = 0$, we get*

$$\mathbb{E}\left[\mathcal{L}\left(\tfrac{1}{T}\sum_{i=1}^T w_i\right)\right] - \mathcal{L}(w^*) \leq \tfrac{RL}{\sqrt{T}} \cdot \sqrt{1 + \tfrac{\sigma^2}{L^2}}.$$

The dependence on the bias is essentially tight, for *any* first order method. Consider the loss function $\mathcal{L}(w) = \frac{1}{2}\alpha w$ defined over the domain $\mathcal{K} = [0, R]$. However, if the gradient oracle is allowed to have a bias of magnitude $\alpha$, it is impossible using any first order algorithm to distinguish between the gradients of $\mathcal{L}$ and that of $\mathcal{L}'(w) := -\frac{1}{2}\alpha w$. Note that both $\mathcal{L}$ and $\mathcal{L}'$ are $L$-Lipschitz for $L \geq \frac{1}{2}\alpha$. While $\mathcal{L}$ is minimized at $w = 0$, $\mathcal{L}'$ is minimized at $w = R$. For any $w \in \mathcal{K}$, $\min\{\mathcal{L}(w) - \mathcal{L}(0), \mathcal{L}'(w) - \mathcal{L}'(R)\} \geq \frac{1}{4}\alpha R$.

Thus, with access to gradients with bias $\alpha$, it is impossible for any first order method to achieve an excess loss that is smaller than $\Omega(\alpha R)$, whereas, with access to gradients with zero bias and variance $\sigma^2$, it is possible to achieve an arbitrarily small excess loss using sufficient number of steps of SGD.

# E  Noisy Label Loss Bound from Discretization

Here we prove Lemma 8, restated below for convenience.

**Lemma 8.** *Let $\mathcal{M}$ be the optimal unbiased randomizer with $\hat{\mathcal{Y}} = [L, U]$, and let $\mathcal{M}_\Delta$ be the optimal unbiased randomizer with $\hat{\mathcal{Y}} = \{L, L + \Delta, \ldots, U - \Delta, U\}$. If $g(\hat{y}, y)$ is $K$-Lipschitz in $\hat{y}$, then $\mathcal{G}(\mathcal{M}_\Delta; \mathcal{P}) \leq \mathcal{G}(\mathcal{M}; \mathcal{P}) + K\Delta$.*

*Proof.* Given any unbiased mechanism $\mathcal{M}$, we construct a mechanism $\mathcal{M}_u$ supported on $\hat{\mathcal{Y}}' := \{L, L + \Delta, \ldots, U - \Delta, U\}$ by applying "unbiased randomized rounding" $\mathrm{URR}_{\hat{\mathcal{Y}}'}$ to the output of $\mathcal{M}$, i.e., $\mathcal{M}_u(y) := \mathrm{URR}_{\hat{\mathcal{Y}}'}(\mathcal{M}(y))$.

For any $\hat{y} \in [L, U]$ such that $\hat{y}_1 \leq \hat{y} \leq \hat{y}_2$ for $\hat{y}_1, \hat{y}_2$ being consecutive elements of $\hat{\mathcal{Y}}'$, $\mathrm{URR}_{\hat{\mathcal{Y}}'}(y)$ returns $\hat{y}'$ drawn as $\hat{y}_1$ with probability $\frac{\hat{y}_2 - \hat{y}}{\hat{y}_2 - \hat{y}_1}$ and $\hat{y}_2$ with probability $\frac{\hat{y} - \hat{y}_1}{\hat{y}_2 - \hat{y}_1}$. Observe that $\mathbb{E}[\mathrm{URR}_{\hat{\mathcal{Y}}'}(\hat{y})] = \hat{y}$ and $|\mathrm{URR}_{\hat{\mathcal{Y}}'}(\hat{y}) - \hat{y}| \leq \Delta$ with probability 1.

Thus, we have that $\mathbb{E}[\mathcal{M}_u(y)] = \mathbb{E}[\mathsf{URR}_{\hat{y}'}(\mathcal{M}(y))] = \mathbb{E}[\mathcal{M}(y)] = y$, and hence $\mathcal{M}_u$ is also unbiased. Moreover,

$$
\begin{aligned}
\mathcal{G}(\mathcal{M}_u; P) = \underset{\substack{y \sim \mathcal{P} \\ \hat{y}' \sim \mathcal{M}_u(y)}}{\mathbb{E}} [g(\hat{y}', y)] &= \underset{\substack{y \sim \mathcal{P} \\ \hat{y} \sim \mathcal{M}(y) \\ \hat{y}' \sim \mathsf{URR}_{\hat{y}'}(\hat{y})}}{\mathbb{E}} [g(\hat{y}', y)] \\
&\leq \underset{\substack{y \sim \mathcal{P} \\ \hat{y} \sim \mathcal{M}(y) \\ \hat{y}' \sim \mathsf{URR}_{\hat{y}'}(\hat{y})}}{\mathbb{E}} [g(\hat{y}, y) + K|\hat{y}' - \hat{y}|] = \mathcal{G}(\mathcal{M}; P) + K\Delta,
\end{aligned}
$$

where the last line follows from $K$-Lipschitzness of $g$. Since $\mathcal{M}_\Delta$ is the optimal unbiased randomizer with output labels $\hat{\mathcal{Y}}'$, we have that $\mathcal{G}(\mathcal{M}_\Delta; \mathcal{P}) \leq \mathcal{G}(\mathcal{M}_u; \mathcal{P}) \leq \mathcal{G}(\mathcal{M}; \mathcal{P}) + K\Delta$. $\qquad\square$

# F   DP Mechanisms Definitions

In this section, we recall the definition of various DP notions that we use throughout the paper.

**Definition 16** (Global Sensitivity). Let $f$ be a function taking as input a dataset and returning as output a vector in $\mathbb{R}^d$. Then, the *global sensitivity* $\Delta(f)$ of $f$ is defined as the maximum, over all pairs $(X, X')$ of adjacent datasets, of $||f(X) - f(X')||_1$.

The (discrete) Laplace distribution with scale parameter $b > 0$ is denoted by $\mathrm{DLap}(b)$. Its probability mass function is given by $p(y) \propto \exp(-|y|/b)$ for any $y \in \mathbb{Z}$.

**Definition 17** (Discrete Laplace Mechanism). Let $f$ be a function taking as input a dataset $X$ and returning as output a vector in $\mathbb{Z}^d$. The *discrete Laplace mechanism* applied to $f$ on input $X$ returns $f(X) + (Y_1, \ldots, Y_d)$ where each $Y_i$ is sampled i.i.d. from $\mathrm{DLap}(\Delta(f)/\varepsilon)$. The output of the mechanism is $\varepsilon$-DP.

Next, recall that the (continuous) Laplace distribution $\mathrm{Lap}(b)$ with scale parameter $b > 0$ has probability density function given by $h(y) \propto \exp(-|y|/b)$ for any $y \in \mathbb{R}$.

**Definition 18** (Continuous Laplace Mechanism, [DMNS06]). Let $f$ be a function taking as input a dataset $X$ and returning as output a vector in $\mathbb{R}^d$. The *continuous Laplace mechanism* applied to $f$ on input $X$ returns $f(X) + (Y_1, \ldots, Y_d)$ where each $Y_i$ is sampled i.i.d. from $\mathrm{Lap}(\Delta(f)/\varepsilon)$. The output of the mechanism is $\varepsilon$-DP.

We next define the discrete and continuous versions of the staircase mechanism [GV14].

**Definition 19** (Discrete Staircase Distribution). Fix $\Delta \geq 2$. The *discrete staircase distribution* is parameterized by an integer $1 \leq r \leq \Delta$ and has probability mass function given by:

$$
p_r(i) = \begin{cases} a(r) & \text{for } 0 \leq i < r, \\ e^{-\varepsilon} a(r) & \text{for } r \leq i < \Delta \\ e^{-k\varepsilon} p_r(i - k\Delta) & \text{for } k\Delta \leq i < (k+1)\Delta \text{ and } k \in \mathbb{N} \\ p_r(-i) \text{ for } i < 0, \end{cases} \tag{3}
$$

where

$$
a(r) =:= \frac{1 - b}{2r + 2b(\Delta - r) - (1 - b)}.
$$

Let $f$ be a function taking as input a dataset $X$ and returning as output a scalar in $\mathbb{Z}$. The *discrete staircase mechanism* applied to $f$ on input $X$ returns $f(X) + Y$ where $Y$ is sampled from the discrete staircase distribution given in (3).

**Definition 20** (Continuous Staircase Distribution). The *continuous staircase distribution* is parameterized by $\gamma \in (0, 1)$ and has probability density function given by:

$$
h_\gamma(x) = \begin{cases} a(\gamma) & \text{for } x \in [0, \gamma\Delta) \\ e^{-\varepsilon} a(\gamma) & \text{for } x \in [\gamma\Delta, \Delta) \\ e^{-k\varepsilon} h_\gamma(x - k\Delta) & \text{for } x \in [k\Delta, (k+1)\Delta) \text{ and } k \in \mathbb{N} \\ h_\gamma(-x) \text{ for } x < 0, \end{cases} \tag{4}
$$

where
$$a(\gamma) =::= \frac{1 - e^{-\varepsilon}}{2\Delta(\gamma + e^{-\varepsilon}(1 - \gamma))}.$$

Let $f$ be a function taking as input a dataset $X$ and returning as output a scalar in $\mathbb{R}$. The *continuous staircase mechanism* applied to $f$ on input $X$ returns $f(X) + Y$ where $Y$ is sampled from the continuous staircase distribution given in (4).

**Definition 21** (Randomized Response, [War65]). Let $\varepsilon \geq 0$, and $q$ be a positive integer. The *randomized response* mechanism with parameters $\varepsilon$ and $q$ (denoted by $\mathsf{RR}_{\varepsilon,q}$) takes as input $y \in \{1, \ldots, q\}$ and returns $\hat{y} \sim \hat{Y}$, where the random variable $\hat{Y}$ is distributed as:
$$\Pr[\hat{y} = \hat{y}] = \begin{cases} \frac{e^\varepsilon}{e^\varepsilon + q - 1} & \text{for } \hat{y} = y \\ \frac{1}{e^\varepsilon + q - 1} & \text{otherwise.} \end{cases} \tag{5}$$
The output of $\mathsf{RR}_{\varepsilon,q}$ is $\varepsilon$-DP.

**Definition 22** (Randomized Response on Bins, [GKK$^+$23]). Let $\varepsilon > 0$, for map $\Phi : \mathcal{Y} \to \hat{\mathcal{Y}}$, $\mathsf{RR\text{-}on\text{-}Bins}_\varepsilon^\Phi$ is defined as the mechanism that on input $y$ samples $\hat{y} \sim \hat{Y}$, where the random variable $\hat{Y}$ is distributed as
$$\Pr[\hat{Y} = \hat{y}] = \begin{cases} \frac{e^\varepsilon}{e^\varepsilon + |\hat{\mathcal{Y}}| - 1} & \text{if } \hat{y} = \Phi(y) \\ \frac{1}{e^\varepsilon + |\hat{\mathcal{Y}}| - 1} & \text{otherwise.} \end{cases}$$

**Definition 23** (Debiased Randomized Response). For $\varepsilon > 0$, the $\varepsilon$-*debiased randomized response* ($\varepsilon$-dbRR) operates by performing randomized response on $(\Phi(y))_{y \in \mathcal{Y}}$, where
$$\Phi(y) = \left( (e^\varepsilon + |\mathcal{Y}| - 1)y - \sum_{y' \in \mathcal{Y}} y' \right) / (e^\varepsilon - 1),$$

namely, the mechanism on input $y$ samples $\hat{y} \sim \hat{Y}$, where the random variable $\hat{Y}$ is distributed as
$$\Pr[\hat{Y} = \hat{y}] = \begin{cases} \frac{e^\varepsilon}{e^\varepsilon + |\mathcal{Y}| - 1} & \text{if } \hat{y} = \Phi(y) \\ \frac{1}{e^\varepsilon + |\mathcal{Y}| - 1} & \text{otherwise.} \end{cases}$$

Finally, we prove Proposition 7, restated below for convenience.

**Proposition 7.** *If $\hat{\mathcal{Y}}$ contains just the smallest and largest values among the outputs of $\varepsilon$-dbRR, the LP in $\mathsf{ComputeOptUnbiasedRand}_\varepsilon(P, \hat{\mathcal{Y}})$ is feasible.*

*Proof.* Let $y_0 = \min(\mathcal{Y})$ and $y_1 = \max(\mathcal{Y})$. The smallest and largest values among the outputs of $\varepsilon$-dbRR are given as
$$\hat{y}_0 = \left( (e^\varepsilon + |\mathcal{Y}| - 1) \cdot y_0 - \sum_{y \in \mathcal{Y}} y \right) / (e^\varepsilon - 1),$$
$$\hat{y}_1 = \left( (e^\varepsilon + |\mathcal{Y}| - 1) \cdot y_1 - \sum_{y \in \mathcal{Y}} y \right) / (e^\varepsilon - 1).$$

Consider the mechanism $\mathcal{M}$ such that for any input $y \in \mathcal{Y}$ it holds that
$$\mathcal{M}(y) = \begin{cases} \hat{y}_0 & \text{w.p. } \frac{\hat{y}_1 - y}{\hat{y}_1 - \hat{y}_0}, \\ \hat{y}_1 & \text{w.p. } \frac{y - \hat{y}_0}{\hat{y}_1 - \hat{y}_0}. \end{cases}$$

Note $\mathcal{M}$ is unbiased, namely $\mathbb{E}[\mathcal{M}(y)] = y$ for all $y \in \mathcal{Y}$. We show
$$\frac{\max_y \Pr[\mathcal{M}(y) = \hat{y}_0]}{\min_y \Pr[\mathcal{M}(y) = \hat{y}_0]} = \frac{\Pr[\mathcal{M}(y_0) = \hat{y}_0]}{\Pr[\mathcal{M}(y_1) = \hat{y}_0]}$$
$$= \frac{(e^\varepsilon + |\mathcal{Y}| - 1) \cdot y_1 - \sum_{y \in \mathcal{Y}} y - (e^\varepsilon - 1)y_0}{(e^\varepsilon + |\mathcal{Y}| - 1) \cdot y_1 - \sum_{y \in \mathcal{Y}} y - (e^\varepsilon - 1)y_1}$$
$$= \frac{\sum_{y \neq y_0}(y_1 - y) + e^\varepsilon(y_1 - y_0)}{\sum_{y \neq y_0}(y_1 - y) + (y_1 - y_0)} \leq e^\varepsilon,$$

where the last step follows because $\sum_{y \neq y_0}(y_1 - y) \geq 0$ and for $a, b, c > 0$ it holds that $(a + b)/(a + c) \leq b/c$. Similarly, it holds that $\max_y \Pr[\mathcal{M}(y) = \hat{y}_1]/\min_y \Pr[\mathcal{M}(y) = \hat{y}_1] \leq e^\varepsilon$. $\qquad\square$

# G Additional Details of Experiments

All our experiments were performed using NVidia P100 GPUs.

## G.1 Hyperparameter Details

**Criteo Sponsored Search Conversion.** We use a neural network that takes as input concatenation of floating point features, as well as categorical features using an embedding table of dimension four for each. The neural network had two hidden layers of dimensions $64$ and $32$ respectively. The training was performed on a random $80\%$ of the dataset using `RMSProp` algorithm using the squared loss objective, with learning rate of $10^{-4}$, $\ell_2$-regularization of $10^{-4}$, batch size of $1,024$ for $50$ epochs. The remaining $20\%$ of the dataset was used to report the test loss.

This architecture choice was chosen by performing a hyperparameter search over various choices, and this choice seemed to give the best results for all the randomizers considered.

**US Census.** We use a neural network that takes as input concatenation of floating point features, as well as categorical features using an embedding table of dimension eight for each. The neural network had two hidden layers of dimensions $128$ and $64$ respectively. The training was performed on a random $80\%$ of the dataset using `RMSProp` algorithm using the squared loss objective, with $\ell_2$-regularization of $10^{-4}$, batch size of $8,192$. A varying learning rate in $\{10^{-2}, 10^{-3}, 10^{-4}\}$ and a varying number of epochs in $\{50, 200\}$ were used, and the best test loss was reported for training with each randomizer. The remaining $20\%$ of the dataset was used to report the test loss.

## G.2 Early Stopping

In training on the AppAds dataset, we found that the training would often be unstable, with the training loss blowing up after several steps, especially for smaller values of $\varepsilon$, which corresponded to faster blow up. We used early stopping, keeping track of intermediate states, to select the best model that minimizes the training loss, and used this to report the test loss.

## G.3 Complexity of Computing the Mechanisms

Even though the wall clock time for computing the optimal unbiased randomizer using an LP solver is significantly larger than that of computing the optimal RR-on-Bins randomizer (for which there is a highly efficient dynamic programming algorithm [GKK+23]), this is still orders of magnitude smaller than that of ML model training. We note that while the noisy label loss decreases as $\Delta$ in Algorithm 2 is made smaller, this causes the computational cost of solving the LP to increase. This trade-off is demonstrated in the following table, which shows the running time of the LP for the unbiased randomizer, the noisy label loss and the final test loss, for different mesh sizes (parameter $n$ in Algorithm 2) for $\varepsilon = 1$ on the US Census dataset we study (prediction of number of weeks worked in $\{1, \ldots, 52\}$).

| Mechanism | Mesh size | Computing mechanism wall-clock time (secs) | Noisy label loss | Test loss |
|---|---|---|---|---|
| RR-on-Bins | n/a | 0.154 | 79.71 | 172.44 |
| Opt. Unbiased | 52 | 2.38 | 1288.21 | 134.44 |
| Opt. Unbiased | 416 | 17.1 | 1275.22 | 134.43 |
| Opt. Unbiased | 1664 | 161 | 1274.71 | 134.43 |

The test loss is quite similar for various mesh sizes, even though the noisy label loss does improve slightly with finer discretization. This suggests that the unbiasedness was key to the improvements over RR-on-Bins and the discretization of the output set does not hurt performance as much.

The computation time increases considerably when the number of input labels is large, e.g., in the Criteo Search Sponsored Conversion Logs dataset, where there were $401$ input labels $\{0, \ldots, 400\}$.

