# OpenReview forum: "Optimal Unbiased Randomizers for Regression with Label Differential Privacy"
_NeurIPS.cc/2023/Conference — NeurIPS 2023 poster_

### Official Review · Reviewer_oAvy · 2023-06-30

**Soundness:** 4 excellent
**Presentation:** 4 excellent
**Contribution:** 3 good
**Rating:** 7
**Confidence:** 4

**Summary:**

This paper investigates the bias of the state-of-the-art label-differential privacy (label-DP) mechanism proposed by Ghazi et al. (2023) and proposes bias-corrected randomizers achieved through minimizing of a constrained linear programming approach. The proposed label-DP mechanisms demonstrate lower mean squared errors empirically by carefully tuning the hyperparameters in deep learning.

**Strengths:**

1. The main result of this paper is intriguing and original. The paper provides a compelling example illustrating the bias present in the state-of-the-art label-DP mechanism and proposes a new mechanism to address this bias. Additionally, the authors thoroughly investigate the empirical performance of the proposed algorithm and show that it outperforms the state-of-the-art.

2. This paper is well-written. The authors effectively communicate their contributions and main findings in a clear and concise manner. Furthermore, the organization of the paper is well-structured.

**Weaknesses:**

1. The proposed algorithm introduces extra computational cost. In order to obtain the unbiased randomizer, a constrained linear programming (LP) problem needs to be solved, which introduces extra computational cost. Although the authors demonstrate the feasibility of the LP, the term "feasible" left me uncertain, and it would be more informative to explicitly state the computational complexity involved.

 2. The authors should consider the potential privacy implications of hyperparameter tuning. In the experimental section, they mention the careful tuning of hyperparameters. However, if the validation set used for this tuning procedure is not label DP, there is a risk of privacy leakage.

**Questions:**

See the weakness part.

**Limitations:**

Yes

---

> ### Author Rebuttal · Authors · 2023-08-09
>
> We thank the reviewer for thoughtful comments and questions. We include below the responses to the questions.
>
> > Computational cost and “feasibility”
>
> Firstly, we clarify that “a linear program (LP) is feasible” (e.g. in Prop. 7) means that the feasible set, a.k.a. the solution space of the LP is non-empty. This is important to show because otherwise, the method is not guaranteed to return a valid $\varepsilon$-DP mechanism.
>
> (Same response as to Reviewer Vhiv) While both the optimal biased and optimal unbiased randomizers can each be written as a solution to an LP, the solution to the former has a special structure that admits a dynamic programming algorithm (RR-on-Bins from prior work).  However, no similar property seems to hold for the solution to the latter; hence explicitly solving the LP is the only available option now. Fortunately, even though solving the LP can be expensive, it is a “one-time” computation in our setting; the training time typically largely dominates the time taken to solve the LP. In addition, the LP also has a “knob”, namely mesh size, that can be used to tradeoff the LP computation and the utility of the unbiased algorithm.
>
> The following Table shows the running time of the LP for the unbiased randomizer, the noisy label loss and the final test loss, for different mesh sizes (parameter $n$ in Algorithm 2) for $\varepsilon = 1$ on the US Census dataset we study (prediction of number of weeks worked in $\{1, \ldots, 52\}$). We note the following:
> * Even though the wall clock time for computing the optimal unbiased randomizer is significantly larger than that of the optimal biased one, this is still orders of magnitude smaller than the ML model training.
> * The test loss is quite similar for various mesh sizes, even though the noisy label loss does improve slightly with finer discretization. This suggests that the unbiasedness was key to the improvements over RR-on-Bins and the discretization of the output set does not affect performance as much.
>
> | **Mechanism** | **Mesh size** | **Computing mechanism wall-clock time** | **Noisy label loss** | **Test loss** |
> |---|---|---|---|---|
> | RR-on-Bins | n/a | 0.154 s | 79.71 | 172.44 |
> | Optimal Unbiased Randomizer | 52 | 2.38 s | 1288.21 | 134.44 |
> | Optimal Unbiased Randomizer | 416 | 17.1 s | 1275.22 | 134.43 |
> | Optimal Unbiased Randomizer | 1664 | 161 s | 1274.71 | 134.43 |
>
> > Privacy implications of hyperparameter tuning
>
> We thank the reviewer for raising this important point. Indeed, hyperparameter tuning in general has additional privacy costs, and how to tune hyperparameters privately and efficiently is an active research topic [1, 2]. Consequently, it is common in the private ML literature to separate the question of private hyperparameter tuning and private training, and focus on comparing the privacy-utility trade-off under optimal hyperparameters [e.g., 3, 4, 5, 6]. In this paper, we follow this convention.
>
> **References:**\
> [1] Nicolas Papernot and Thomas Steinke.  Hyperparameter tuning with Renyi differential privacy, 2021.\
> [2] Sander, Tom, Pierre Stock, and Alexandre Sablayrolles. Tan without a burn: Scaling laws of DP-SGD, 2023.\
> [3] Malek, Mani, et al. Antipodes of label differential privacy: PATE and ALIBI, 2021.\
> [4] He, J., Li, X., Yu, D., Zhang, H., Kulkarni, J., Lee, Y. T., Backurs, A., Yu, N., & Bian, J. Exploring the Limits of Differentially Private Deep Learning with Group-wise Clipping, 2022.\
> [5] Kurakin, A., Song, S., Chien, S., Geambasu, R., Terzis, A., & Thakurta, A. Toward Training at ImageNet Scale with Differential Privacy, 2022.\
> [6] De, S., Berrada, L., Hayes, J., Smith, S. L., & Balle, B. Unlocking High-Accuracy Differentially Private Image Classification through Scale, 2022.

---

> > ### Comment · Reviewer_oAvy · 2023-08-15
> >
> > Thank you for your detailed response. I would like to keep my score and suggest acceptance.

---

### Official Review · Reviewer_YHYY · 2023-07-06

**Soundness:** 3 good
**Presentation:** 4 excellent
**Contribution:** 2 fair
**Rating:** 7
**Confidence:** 4

**Summary:**

This paper proposes a differentially private algorithm for regression problems. The algorithm protects the privacy of the labels ("label DP"), in contrast to the entire example. The canonical application of this is digital advertising, where the label might be transaction data from a separate website. Furthermore, the algorithm operates in what [GKK+23] term the "feature-oblivious" model, where a private algorithm, operating solely on the labels, sends a message to the "features party," who then uses these together to learn a model.

[GKK+23] studied a label randomizer that aims to minimize the difference between the true and noisy labels. This work studies a unbiased randomizers. In addition to experiments, the paper provides some theoretical evidence that unbiased estimators, even if they have high variance, may be superior.

The private algorithm (which, again, only has access to the labels) proceeds in two steps. First, it uses a private histogram to estimate a prior over the labels. It then solves a linear program, the output of which is our label randomizer. The LP minimizes the expected difference between the true and noisy labels, subject to privacy, unbiasedness, and normalization constraints.

An important technical question is the randomizer's output space. Let $\mathcal{Y}\subseteq \mathbb{R}$ be the label space and $\hat{\mathcal{Y}}\subseteq \mathbb{R}$ the set of possible outputs of the randomizer. The solution to the LP has $|\mathcal{Y}|\times|\hat{\mathcal{Y}}|$ entries, representing a probability distribution over $\hat{\mathcal{Y}}$ for each entry in $\mathcal{Y}$. Thus it is clear that this approach demands both sets be finite (and of modest size, to solve the LP). What is less clear, due to the unbiasedness constraint, is what $\hat{\mathcal{Y}}$ should be. Note that if $\min \mathcal{Y} =0 $ and $\hat{\mathcal{Y}}=\mathcal{Y}$, then unbiasedness demands we map $0\to 0$ with probability 1, violating privacy. Thus the endpoints of $\hat{\mathcal{Y}}$ must exceed those of $\mathcal{Y}$. The authors set the endpoints of $\hat{\mathcal{Y}}$ so that (provably) the LP is feasible. They "fill in" the rest of $\hat{\mathcal{Y}}$ with a grid, as finely spaced as their computational constraints allow.

For a fixed $\hat{\mathcal{Y}}$ and prior on $\mathcal{Y}$, the LP finds the unbiased private randomizer with the lowest expected difference (or loss) in labels. Experiments demonstrate improvements over prior work on three data sets.

I noticed that parts of the submission (introducing the label DP recipe, describing data sets, and reviewing related work) had substantial overlap with text from GKK+23. After discussion with my chair, I have not used this information in my evaluation of the paper.

**Strengths:**

Without unbiasedness constraints, GKK+23 find the best randomizer is of a form they call "RR-on-Bins." The authors nicely sum up their core innovation: "the addition of an unbiasedness constraint to the linear program leads to solutions to the LP that (i) are not RR-on-Bins solutions, (ii) can have substantially higher variance than RR-on-Bins, and (iii) nevertheless have a much lower train and test error due to the reduction in bias." This is a clear observation that allows them to move past prior work.

I buy that this is a practical problem, that the feature-oblivious model is useful.

Their approach does seem to significantly improve on that of GKK+23 on the data sets they tested. Their experimental results are complemented by their formal claims (Theorem 6 and Proposition 7) about the output label space of randomizers.

With minor exceptions, I found the presentation very clear throughout.

**Weaknesses:**

In light of the work of GKK+23, this paper has limited novelty. Originality is also a serious weakness: GKK+23 point out that "our noising mechanism typically introduces a bias; mitigating it, say by adding unbiasedness constraints into the LP, is an interesting direction."

The title says the work finds "optimal" unbiased randomizers, but that optimality only holds once the choice of output space is selected. Theorem 6 (there is an optimal unbiased randomizer with at most $2|\mathcal{Y}|$ labels) and Proposition 7 (selecting endpoints to ensure feasibility) are good steps, but do not get us all the way there. As far as I can tell, it remains possible that some other scheme generates better unbiased randomizers, at least for some priors.

The paper's focus is on feature-oblivious algorithms, but I would have liked to see a comparison with other label-DP algorithms that solve this problem. How much are we giving up by moving to feature-oblivious?

**Questions:**

Is there evidence (even informal) that this discretization approach is close to optimal? For instance, can we rule out that the LP's objective is highly sensitive to the choice of endpoints (in the regime considered)?

Can you briefly sketch out what you feel are the key innovations beyond GKK+23?

Estimating the prior requires some privacy budget for each bin. Do you expect that, as $|\mathcal{Y}|$ grows, another approach will perform better than yours?

**Limitations:**

The algorithm operates in the feature-oblivious model; other label-DP algorithms might perform better in different settings.

The algorithm assumes prior knowledge of $\mathcal{Y}$; it is not always clear where this information comes from.

---

> ### Author Rebuttal · Authors · 2023-08-09
>
> We thank the reviewer for thoughtful comments and questions. We include below the responses to the questions.
>
> > key innovations beyond GKK+23
>
> GKK+23 proposed an optimal (biased) randomizer (RR-on-Bins). While an unbiased mechanism was mentioned as a future direction in GKK+23, it was not clear how to realize this in practice. The key innovations of our paper beyond GKK+23 include a formulation and theoretical justification of an unbiased randomizer, a practical algorithm based on a heuristic discretization approach to the LP, a systematic evaluation with optimal hyperparameters and empirical results that substantially outperform the previous state-of-the-art.
>
> > discretization approach is close to optimal?
>
> Thank you for the important question. Firstly, Theorem 6 implies the output set is finite and hence some boundary exists. The way we constructed the heuristic for choosing the boundary points for the output set was by experimenting with various strategies, first ensuring feasibility of the LP, and then expanding it and stopping when we start seeing diminishing returns on the LP objective. We also empirically observe that the final mechanism is supported on values that are away from the boundaries we consider, so it seems likely that expanding the boundaries further does not change the optimal solution.
>
> The mesh size controls a trade-off between computation time for solving the LP and the value of noisy label loss. We note that it is possible to bound the optimality gap due to discretization in terms of Lipschitz constant of the loss and width of the mesh discretization interval (see Lemmas towards the end of this rebuttal). In practice however, (see the table in response to Reviewer Vhiv), we find that a smaller mesh size is already able to recover good test loss, especially for small $\varepsilon$ values, even though a finer discretization reduces the noisy label loss by a small amount.
>
> We leave it to future work for a more principled method to compute the optimal unbiased randomizer. In practice, we find this heuristic to be sufficient for extracting utility out of these unbiased randomizers. We will add more discussion on this in the revision.
>
> > Estimating the prior requires some privacy budget for each bin ... as |Y| grows, another approach will perform better than yours?
>
> Note that, in estimating the prior, the amount of noise added to each bin is sampled from $\mathrm{Lap}(2/\varepsilon_1)$, since changing one label can modify the counts for only two bins (reducing one and increasing the other). Thus, we do not need a privacy budget for each bin.
>
> > feature-oblivious algorithms ... comparison with other label-DP algorithms that solve this problem. How much are we giving up by moving to feature-oblivious?
>
> Thanks for the suggestion! While there are feature-aware LabelDP algorithms for classification problems, we are not aware of any existing feature-aware LabelDP algorithms for regression problems. In this paper, we focus on feature-oblivious algorithms for simplicity. But we note feature-awareness is an orthogonal component that can be easily added by extending our algorithm to use feature-aware priors instead of a global prior. One natural approach mentioned in the Conclusion section is to cluster the  input features and build separate priors for each cluster. With this extension, the utility of our algorithm could potentially be improved -- but the trade-off between utility gain and implementation complexity will be heavily task and data dependent. We leave a systematic study to the future work.
>
> ---
>
> **Lemma:** Let $M$ be the optimal unbiased randomizer with output labels bounded in $[A, B]$. Let $M_d$ be the optimal unbiased mechanism with output labels in $\{A, A+d, A+2d, …, B-d, B\}$. Suppose for all input labels $x$, output labels $y \in [A, B]$ and $c \in [-d, d]$ such that $y+c \in [A,B]$ and $|\ell(x, y) - \ell(x, y+c)| < C_d$. Then the noisy label loss $\mathcal{G}(M_d; \mathcal{P}) \leq \mathcal{G}(M; \mathcal{P}) + C_d$.
>
> _Proof._ From $M$, we construct an unbiased mechanism supported on $\{A, A+d, A+2d, …, B-d, B\}$ by a postprocessing step called ‘unbiased rounding’. For an output label $k \in [A+nd, A+(n+1)d]$, any time we see $k$ as an output of $M$, we instead emit the value $A+nd$ with probability $(k - (A+nd)) / d$, and emit $A+n(d+1)$ with probability $(A+(n+1)d - k) / d$. This post processing step is unbiased, and therefore preserves the mechanism being unbiased. It is clearly supported on $\{A, A+d, A+2d, …, B-d, B\}$. Lastly, because $|\ell(x, y) - \ell(x, y+c)| < C_d$, unbiased rounding at the output value $k$ increases the noisy label loss at most by $\Pr(output=k)*C_d$. Summing over all possible output labels $k$, we get $\mathcal{G}(M_u; \mathcal{P}) \leq \mathcal{G}(M; \mathcal{P}) + C_d$ where $M_u$ is the mechanism created by performing unbiased rounding on $M$. Since $M_d$ is the optimal mechanism over a set that includes $M_u$, we get $\mathcal{G}(M_d; \mathcal{P}) - \mathcal{G}(M; \mathcal{P}) < C_d$.
>
> **Lemma:** Suppose $\ell(x,y)$ is $K$-Lipschitz for $x,y \in [A, B]$. Then $C_d < Kd$ in the lemma above.
>
> _Proof._ By the $K$-Lipschitzness of $\ell$, $|\ell(x, y) - \ell(x, y+c)| < K|c|$, and $c \in [-d, d]$, giving the lemma.
>
> Note that for the case of $\ell(x,y) = \frac{1}{2} (x-y)^2$, $\ell$ is $(B-A)$-Lipschitz. These two lemmas taken together show that as $d$ goes to $0$, that is, the mesh gets finer, the noisy label loss of the unbiased mechanism obtained from the LP approaches the optimal noisy label loss, and gives a bound on the excess loss in terms of $d$.

---

> > ### Comment · Reviewer_YHYY · 2023-08-12
> >
> > Thanks for the detailed response. Your elaboration on the discretization makes me feel much better about the approach. I will increase my score to a 7.
> >
> > On estimating the prior: of course you are correct, apologies for the mistake. If $|\mathcal{Y}|$ is large relative to the number of samples, our estimate of the prior might be poor. I doubt this is of much concern; as you mention elsewhere, a different discretization or grouping of labels would likely work well. (No need to reply to this comment.)

---

### Official Review · Reviewer_DWLv · 2023-07-07

**Soundness:** 3 good
**Presentation:** 3 good
**Contribution:** 3 good
**Rating:** 6
**Confidence:** 3

**Summary:**

The paper studies the regression problems under label differential privacy (DP). It proposes a novel randomizer that generates high-quality DP labels which can be used to train a regressor. The proposed randomized mechanism is sound and the experiments on various benchmark datasets and different privacy budgets demonstrated the improvements over the baselines.

Overall I think the contribution of the paper is significant, but the proposed mechanism seems to be limited to the case where the labels are discrete (not continuous). Based on that I recommend a weak accept for this paper.

**Strengths:**

1. The contribution is significant and the idea is novel. The paper proposes a novel randomized mechanism that can generate unbiased DP labels which helps to train an unbiased regressor.

2. The paper conducts extensive experiments on different benchmark datasets and different epsilon and shows that a noticeable reduction of mean squared error  when using the proposed mechanism over the previous works.



**Weaknesses:**

1. The paper focuses on optimal mechanism for epsilon-DP and this topic is pretty studied in some previous work such as  the stair-case mechanism. It would be great if there is any discussion about extension to (epsilon, delta)-DP.

2. The settings are limited to the discrete (but not continuous) choice of labels which only applies for ordinal regression.

3. The authors did not attach the codes so it might be hard to reproduce the experiments.

**Questions:**

Can the authors explain more why the proposed mechanism is better than the optimal stair-case mechanism for example if we look at Figure 4.a , especially even when epsilon is large? I used the staircase mechanism before and this turned out to be better than many baseline mechanisms when epsilon is large enough.


**Limitations:**

The paper  discusses clearly the limitations of their proposed mechanism compared to the baselines in the Conclusion section.

---

> ### Author Rebuttal · Authors · 2023-08-09
>
> We thank the reviewer for thoughtful comments and questions. We include below the responses to the questions.
>
> > Comparison of “optimal unbiased randomizers” to the staircase mechanism.
>
> Indeed, the staircase mechanism was introduced as the optimal noise mechanism minimizing the _worst case error_, namely the error for any true value, and moreover the domain of the true value is unbounded, namely all of $\mathbb{R}$. Our optimization problem is different in that we are optimizing for the _average_ squared loss between the noisy and true labels and we assume that our domain is bounded. Note that the staircase mechanism is usually applied in the central model of DP, where the noise is added to an aggregate value (which can be unbounded, but has bounded sensitivity), whereas, in our setting, we are applying the staircase mechanism in the local model of DP.
>
> > It would be great if there is any discussion about extension to (epsilon, delta)-DP.
>
> For our specific approach, it does not seem that relaxing to $(\varepsilon, \delta)$-DP will be beneficial. For the first stage of estimating the histogram privately, one could potentially use an approximate- DP mechanism, but we believe that is not likely to change the prior significantly. For the second stage of randomizing labels, it is known that approximate-DP may not be helpful in the local model [1]. Thank you for raising this question; we will add a discussion about this in the Conclusions section.
>
> > The settings are limited to the discrete (but not continuous) choice of labels which only applies for ordinal regression.
>
> We can apply our techniques even to the continuous case by discretizing the domain (such as in the experiments with the Criteo Sponsored Search Conversion Log Dataset, where the `SalesAmountInEuro` field was an arbitrary floating point number). There is indeed some utility loss due to the discretization, but it should not be significant, as it can be upper bounded in terms of Lipschitz constant of the loss and the width of the discretization interval. Moreover, the rounding can be done in a randomized manner that preserves the expectation (e.g., 0.3 rounds to 1 w.p. 0.3 and to 0 w.p. 0.7), so the final noisy label as a result of rounding and applying the randomizer is unbiased. (Please see the Lemmas in response to Reviewer YHYY for a similar argument about discretization of the output labels.)
>
> **References**\
> [1] Bun, Nelson, Stemmer. Heavy Hitters and the Structure of Local Privacy, PODS 2018.

---

### Official Review · Reviewer_Vhiv · 2023-07-27

**Soundness:** 3 good
**Presentation:** 4 excellent
**Contribution:** 3 good
**Rating:** 7
**Confidence:** 3

**Summary:**

This paper proposes a new family of DP label randomizers for regression models. They show that these randomizers improve the MSE of the training set at the expense of the noisy label loss, indicating an alternate bias-variance tradeoff to other similar works.

**Strengths:**

The strength of this paper is the theoretical and empirical evidence of the randomizer. It is evident that it outperforms the unbiased randomizers by significant margins in the test error, but this comes at the cost of the noisy label loss.

**Weaknesses:**

However, it would have been helpful to include what this tradeoff means practically and its implications.

**Questions:**

Overall, I think the paper is well-written and has enough experiments to validate its conjectures. The novelty of the paper is in its theoretical results and improved guarantees.

Some points/comments that could be improved are:
$\hat{y}$, the noisy label, is never defined in Section 3?
The end of Section 3, particularly the observations for $\epsilon$ are a bit unwieldy to read -- this would probably be better presented with a table or figure
Why is the computational complexity not compared between the unbiased and biased randomizers? This would help understand what tradeoffs of this method are
It would be nice to see an analysis of why the method works better on one dataset versus the other
In Section 4.3, it is mentioned that for smaller $\epsilon$’s one can see the standard error increase -- is this only true for these types of algorithms or simpler models as well?


**Limitations:**

Yes

---

> ### Author Rebuttal · Authors · 2023-08-09
>
> We thank the reviewer for the positive feedback and useful suggestions. We will add the definition of $\hat{y}$ and incorporate other suggested changes in a future revision.
>
> > Trade-off between noisy label loss and test loss
>
> Our main goal in the paper is to minimize the final test loss. The noisy label is only an intermediate metric of interest. Note that the noisy label loss decomposes as bias and variance. The main observation of this paper is that _reducing the bias, even at the cost of blowing up the variance (by orders of magnitude) is beneficial for the end goal of minimizing the test loss_. This seems to be consistent with our experiments, where for mechanisms without bias (unclipped Laplace & staircase mechanisms, and our optimal unbiased randomizer), we find that reducing the variance corresponds to reducing the test loss as well.
>
> > Computational complexity
>
> (Same response as to Reviewer oAvy) While both the optimal biased and optimal unbiased randomizers can each be written as a solution to an LP, the solution to the former has a special structure that admits a dynamic programming algorithm (RR-on-Bins from prior work).  However, no similar property seems to hold for the solution to the latter; hence explicitly solving the LP is the only available option now. Fortunately, even though solving the LP can be expensive, it is a “one-time” computation in our setting; and the training time typically largely dominates the time taken to solve the LP. In addition, the LP also has a “knob”, namely mesh size, that can be used to tradeoff the LP computation and the utility of the unbiased algorithm.
>
> The following Table shows the running time of the LP for the unbiased randomizer, the noisy label loss and the final test loss, for different mesh sizes (parameter $n$ in Algorithm 2) for $\varepsilon = 1$ on the US Census dataset we study (prediction of number of weeks worked in $\{1, \ldots, 52\}$). We note the following:
> * Even though the wall clock time for computing the optimal unbiased randomizer is significantly larger than that of the optimal biased one, this is still orders of magnitude smaller than the ML model training.
> * The test loss is quite similar for various mesh sizes, even though the noisy label loss does improve slightly with finer discretization. This suggests that the unbiasedness was key to the improvements over RR-on-Bins and the discretization of the output set does not affect performance as much.
>
> | **Mechanism** | **Mesh size** | **Computing mechanism wall-clock time** | **Noisy label loss** | **Test loss** |
> |---|---|---|---|---|
> | RR-on-Bins | n/a | 0.154 s | 79.71 | 172.44 |
> | Optimal Unbiased Randomizer | 52 | 2.38 s | 1288.21 | 134.44 |
> | Optimal Unbiased Randomizer | 416 | 17.1 s | 1275.22 | 134.43 |
> | Optimal Unbiased Randomizer | 1664 | 161 s | 1274.71 | 134.43 |
>
> > In Section 4.3, it is mentioned that for smaller $\varepsilon$’s one can see the standard error increase -- is this only true for these types of algorithms or simpler models as well?
>
> We conjecture that this might be data dependent (e.g., specific to the AppAds dataset).  Note that the increase in standard error bars is evident even for other unbiased mechanisms such as Laplace and Staircase mechanisms as well.

---

### Decision · Program_Chairs · 2023-09-21

**Decision:**

Accept (poster)

**Comment:**

The authors were unanimous in identifying the merits of this paper, and its fit for publication at NeurIPS.

One of the reviewers identified several passages in the paper that were nearly identical to the arxiv version of GKK+23, which was published at ICLR. Before publication, the authors MUST do an editing pass to remove any copied text to avoid plagiarism.